# The CHD6 chromatin remodeler is an oxidative DNA damage response factor

Shaun Moore[1], N. Daniel Berger[1], Martijn S. Luijsterburg[2], Cortt G. Piett[3], Fintan K.T. Stanley[1], Christoph U. Schräder[1], Shujuan Fang[1], Jennifer A. Chan[1], David C. Schriemer[1], Zachary D. Nagel[3], Haico van Attikum[2] & Aaron A. Goodarzi[1]

Cell survival after oxidative DNA damage requires signaling, repair and transcriptional events often enabled by nucleosome displacement, exchange or removal by chromatin remodeling enzymes. Here, we show that Chromodomain Helicase DNA-binding protein 6 (CHD6), distinct to other CHD enzymes, is stabilized during oxidative stress via reduced degradation. CHD6 relocates rapidly to DNA damage in a manner dependent upon oxidative lesions and a conserved N-terminal poly(ADP-ribose)-dependent recruitment motif, with later retention requiring the double chromodomain and central core. CHD6 ablation increases reactive oxygen species persistence and impairs anti-oxidant transcriptional responses, leading to elevated DNA breakage and poly(ADP-ribose) induction that cannot be rescued by catalytic or double chromodomain mutants. Despite no overt epigenetic or DNA repair abnormalities, CHD6 loss leads to impaired cell survival after chronic oxidative stress, abnormal chromatin relaxation, amplified DNA damage signaling and checkpoint hypersensitivity. We suggest that CHD6 is a key regulator of the oxidative DNA damage response.

[1] Robson DNA Science Centre, Arnie Charbonneau Cancer Institute, Departments of Biochemistry & Molecular Biology and/or Oncology, Cumming School of Medicine, University of Calgary, Calgary, AB T2N 4N1, Canada. [2] Department of Human Genetics, Leiden University Medical Center, Einthovenweg 20, 2333 ZC Leiden, The Netherlands. [3] Harvard University, School of Public Health, Boston, MA 02115, USA. These authors contributed equally: N. Daniel Berger, Martijn S. Luijsterburg, Cortt G. Piett, Fintan K. T. Stanley. Correspondence and requests for materials should be addressed to A.A.G. (email: A.Goodarzi@ucalgary.ca)

Eukaryotic DNA is packaged with histones to form nucleosomes, the functional unit of chromatin. Chromatin remodeling enzymes adjust nucleosome spacing to regulate DNA accessibility and transcription in response to stimuli, and are essential components of DNA damage responses that are altered often in cancer[1]. The nine-member family of chromodomain-helicase-DNA binding (CHD) chromatin remodeling enzymes is characterized by the double chromodomain and a central ATPase-helicase domain that confers nucleosome respacing, removal or exchange activity[2]. Among the CHD enzymes, CHD1, CHD2, CHD3.1 and CHD4 have roles in DNA damage repair pathways[3–13]. No definitive roles for CHD5–9 within the DNA damage response have been described to date, although both CHD5 and CHD6 expression are known to be altered in cancer[1]. CHD6 was actually the fifth CHD protein discovered and, after initially being called CHD5[14], its name was changed to CHD6 when another protein (now called CHD5) was identified with greater homology to CHD3 and CHD4[15].

CHD chromatin remodelers generally adjust linker DNA length between nucleosomes, increasing per capita histone occupancy and disfavoring sequence-positioned nucleosome deposition[16]. Purified CHD6 disrupts and re-positions nucleosomes, albeit in a non-sliding manner distinct to CHD7/8[17]. CHD6 is expressed ubiquitously in mammalian tissue, although little is known about its molecular function. H3$^{K4me3}$-independent promoter occupancy in mouse embryonic stem cells indicates that CHD6 is present at both sites of active and inactive transcription[18]. The impact of CHD6 enzymatic activity in vivo, as well as what genomic regions are regulated by CHD6 in a human and/or differentiated cell, are still unclear. Catalytic inactivation deletion of exon 12 (encoding a conserved portion of ATPase domain) in mice causes cerebellar defects and ataxia[19]. Morphological analysis revealed no structural cerebellar defects in CHD6 Δexon12 mice, suggesting that CHD6 is not dominant in development, but prevents progressive cerebellar degeneration, which is a frequently documented consequence of failure to suppress or resolve DNA damage leading to neuronal death within the central nervous system[20,21]. CHD6 is a known cancer driver[22] and, according to The Cancer Genome Atlas, is overexpressed in cancers arising in oxidatively stressed tissue microenvironments, including colorectal, uterine, gastric, lung and pancreatic cancers[23,24]. In A549 human lung carcinoma cells, CHD6 messenger RNA (mRNA) has been reported to increase slightly with very low (but not high) ionizing radiation (IR) doses[25], although the significance of this is unclear.

Here, we present evidence that CHD6 stabilizes during oxidative stress, relocates dynamically to sites of oxidative DNA damage and is a key component of the signaling and transcriptional response to reactive oxygen species (ROS) exposure. We define a mechanism by which cell survival in oxidatively stressed human cells is driven by the chromatin remodeler CHD6.

## Results

**CHD6 levels fluctuate and stabilize during oxidative stress**. We first analyzed CHD6 protein levels in cells under various oxidative stress conditions, as previous work had only examined CHD6 mRNA by northern blot after IR[25]. Whole cell extracts of A549 cells grown to confluence in either 3% or ambient (21%) $O_2$ were immunoblotted for CHD6, p53 (as a control for DNA damage response activation) and actin (Fig. 1a). CHD6 was weakly detectable in cells grown in 3% $O_2$, but increased substantially in 21% $O_2$. CHD6 protein levels responded dynamically to $O_2$, and dropped within several hours of cells being transferred from 21 to 3% $O_2$. We also examined whether CHD1, CHD2, CHD3.1 or CHD4, the CHD enzymes previously described to have a role in

the DNA damage response, displayed dynamic alteration in expression under the same conditions (Fig. 1b, c). In all cases, no significant increase in protein expression was observed, highlighting a unique mode of regulation for CHD6 among these enzymes. In contrast to previous reports[25], quantitative PCR (qPCR) revealed no significant increase in CHD6 mRNA over a 24 h period following oxidative stress (Fig. 1d). While etoposide treatment, a source of non-oxidative DNA damage, did not alter CHD6 expression (Fig. 1e), acute $H_2O_2$/IR exposure triggered an increase in total CHD6 protein, indicating a selective response to multiple sources of oxidative stress (Fig. 1f, g). We speculated that the rapid (<30 min) change in CHD6 protein level was likely via reduced proteasome degradation. Fitting with this hypothesis, bortezomib, which blocks proteasome-mediated degradation, stabilized CHD6 levels to those observed after $H_2O_2$/IR (Fig. 1f, g).

**CHD6 is recruited dynamically to oxidative DNA damage**. We next asked whether, in addition to protein level stabilization, oxidative DNA damage alters CHD6 intracellular localization. We cloned human CHD6 complementary DNA (cDNA) into N-terminally green fluorescent protein (GFP)- or C-terminally FLAG-tag expression constructs (controlling for type of epitope and position) and transfected these into A549 cells. CHD6$^{GFP}$ and CHD6$^{FLAG}$ expressed well, did not impact short-term cell viability adversely and were exclusively nuclear (Fig. 1h); moreover, we were also able to detect endogenous CHD6 (Fig. 1h). Bromodeoxyuridine (BrdU)-treated cells were subjected to 355 nm laser micro-irradiation, harvested 5 min later and immunostained for DNA damage response markers. Endogenous CHD6, CHD6$^{GFP}$ and CHD6$^{FLAG}$ localized to micro-irradiation tracks overlapping with γH2AX and/or poly(ADP-ribose) (PAR) (Fig. 1h). Our micro-irradiation power was equivalent to ~8 Gy IR (Supplementary Fig. 1a). CHD6$^{GFP}$ relocalizes to sites of DNA damage in a multi-phasic fashion, involving very fast (~1 min) recruitment, slower accumulation between 1 and 8 min, relatively stable retention from 8 to 16 min and then a steady decline. This is in contrast to XRCC1$^{GFP}$ and PARP1$^{GFP}$ that, under identical conditions, display only rapid recruitment and immediate dispersal (Fig. 1i). Laser micro-irradiation produces many oxidative DNA lesions, including oxidized bases, single-strand breaks (SSBs) and double-strand breaks (DSBs). To better distinguish the general class of DNA damage that attracts CHD6, we used a site-specific system[26] whereby DNA damage is produced at a single genomic locus (a LacO array) by either LacR-fused Fok1 nuclease (DSBs, non-oxidative damage) or LacR-fused KillerRed (ROS-mediated damage) (Fig. 2a). CHD6$^{GFP}$ was not recruited to Fok1 nuclease-induced DNA damage, despite γH2AX formation and XRCC4 retention at the Fok1$^{LacR}$-bound array, indicative of DSB formation and active repair complex formation (Fig. 2b, d). In contrast, CHD6$^{GFP}$ accumulated robustly at sites of ROS-induced DNA damage mediated by light-stimulated KillerRed$^{LacR}$ (Fig. 2c, d). KillerRed-induced DNA damage sites were demarcated by XRCC1 (a key base excision and SSB repair factor), PAR, XRCC4 and weak γH2AX (Fig. 2c–e). KillerRed-mediated DNA damage induction was comparable to 1 mM $H_2O_2$ treatment in media (Fig. 2f–h). These data suggest that ROS-induced DNA lesions, but not enzymatic (non-oxidative) DNA damage types, are required to elicit CHD6 recruitment, similar to our earlier observations with CHD6 expression.

**CHD6 recruitment to DNA damage is PAR dependent**. To address how CHD6 is recruited to DNA damage, we used micro-irradiation, which permits real-time spatiotemporal kinetic analysis of CHD6 relocalization. CHD6 accumulation at DNA

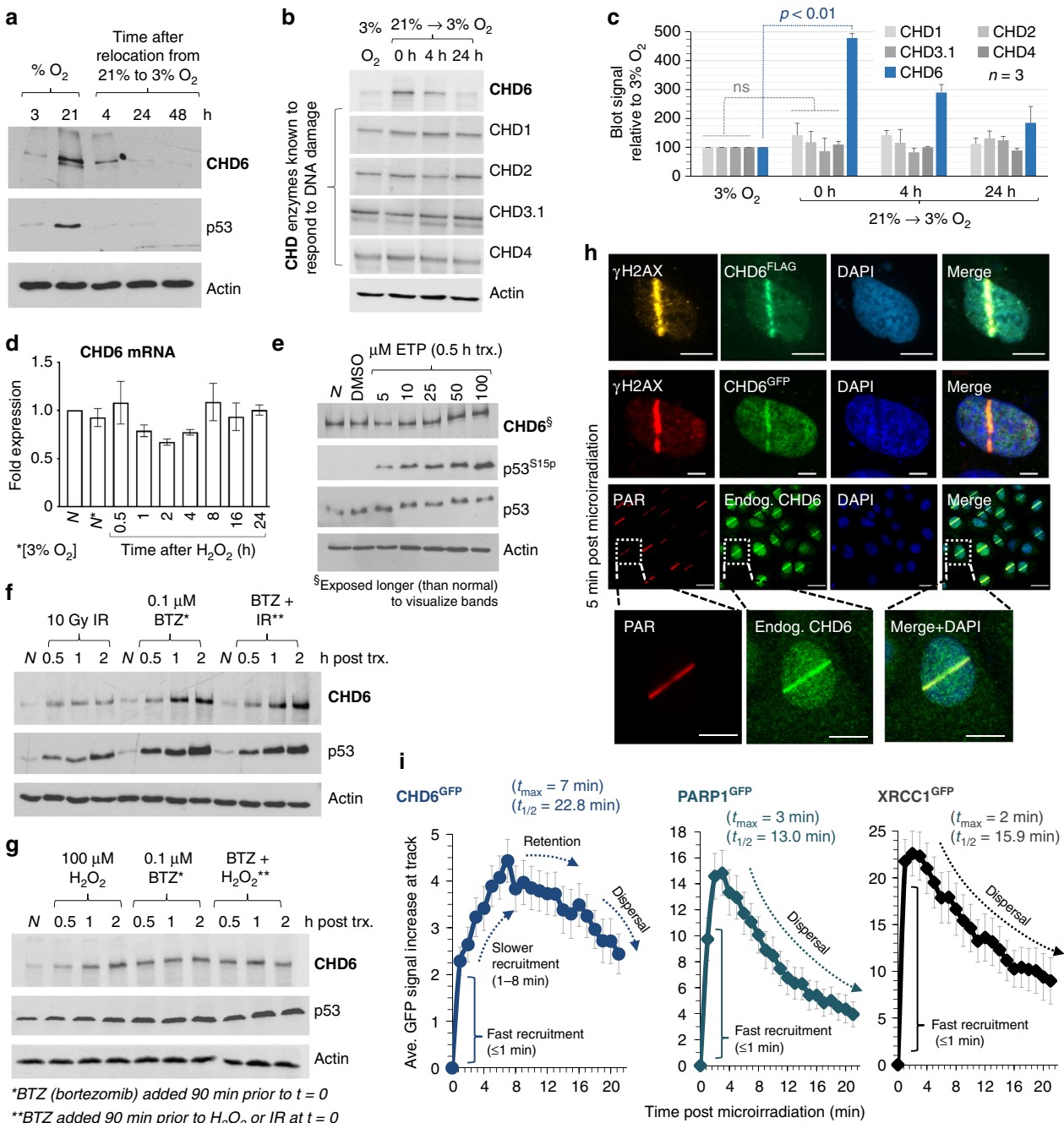

**Fig. 1** CHD6 protein levels and localization respond dynamically to oxidative DNA damage. **a** A549 were cultured at the indicated $O_2$ levels and immunoblotted for CHD6, p53 and actin. Representative blot shown, $n = 3$. **b** Cell extracts from (**a**) were immunoblotted for CHD1, CHD2, CHD3.1, CHD4 or CHD6 and actin. Representative blot shown. **c** Data from (**b**) was quantified, error bars = s.d.; $n = 3$. P values are for Student's t-test. **d** A549 were exposed ± 500 μM $H_2O_2$ over 24 h, before extraction and analysis by qPCR for CHD6 mRNA, error bars = s.e.m.; $n = 3$. **e** A549 were exposed to 5–100 μM etoposide (ETP) for 0.5 h and immunoblotted as in (**a**) with the addition of p53$^{S15p}$. Representative blot shown, $n = 3$. **f**, **g** A549 were exposed to $H_2O_2$ or 10 Gy ionizing radiation (IR)±bortezomib (BTZ) as indicated and immunoblotted as in (**a**). Representative blots shown, $n = 3$. **h** A549 transfected ±CHD6$^{GFP/FLAG}$ microirradiated, fixed 5 min later and immunostained for indicated epitopes; scale bars equal 5 μm. Lower panels show an expanded image of endogenous CHD6 localized at DNA damage tracks; scale bars equal 20 and 5 μm. **i** A549 expressing XRCC1$^{GFP}$, PARP1$^{GFP}$ or CHD6$^{GFP}$ were treated as in (**h**), live imaged over 21 min and quantified over time as described in Methods. $t_{max}$ = average time to maximum signal. $t_{1/2}$ = average time to 50% loss of maximum signal. Error bars = s.e.m.; $n = 30$ cells. In all cases, ±refers to with and without

damage tracks was not affected by small molecule inhibition of the DSB signaling protein kinases ataxia telangiectasia mutated (ATM) and/or DNA-dependent protein kinase (DNA-PK; Supplementary Figs. S1b, c). However, inhibition of PAR polymerase (PARP) activity using Olaparib (PARPi) ablated endogenous CHD6$^{GFP}$ accumulation (Fig. 3a, b, Supplementary Fig. 1d).

These results were confirmed by depletion of PARP1 and PARP2 by small interfering RNA (siRNA; Fig. 3c, Supplementary Figs. 1e, f). PAR chains are degraded by the PAR glycohydrolase (PARG) and loss or inhibition of PARG activity prolongs PAR longevity, as well as the occupancy of PAR-binding proteins[27]. Endogenous CHD6$^{GFP}$ occupancy at micro-irradiation tracks was prolonged

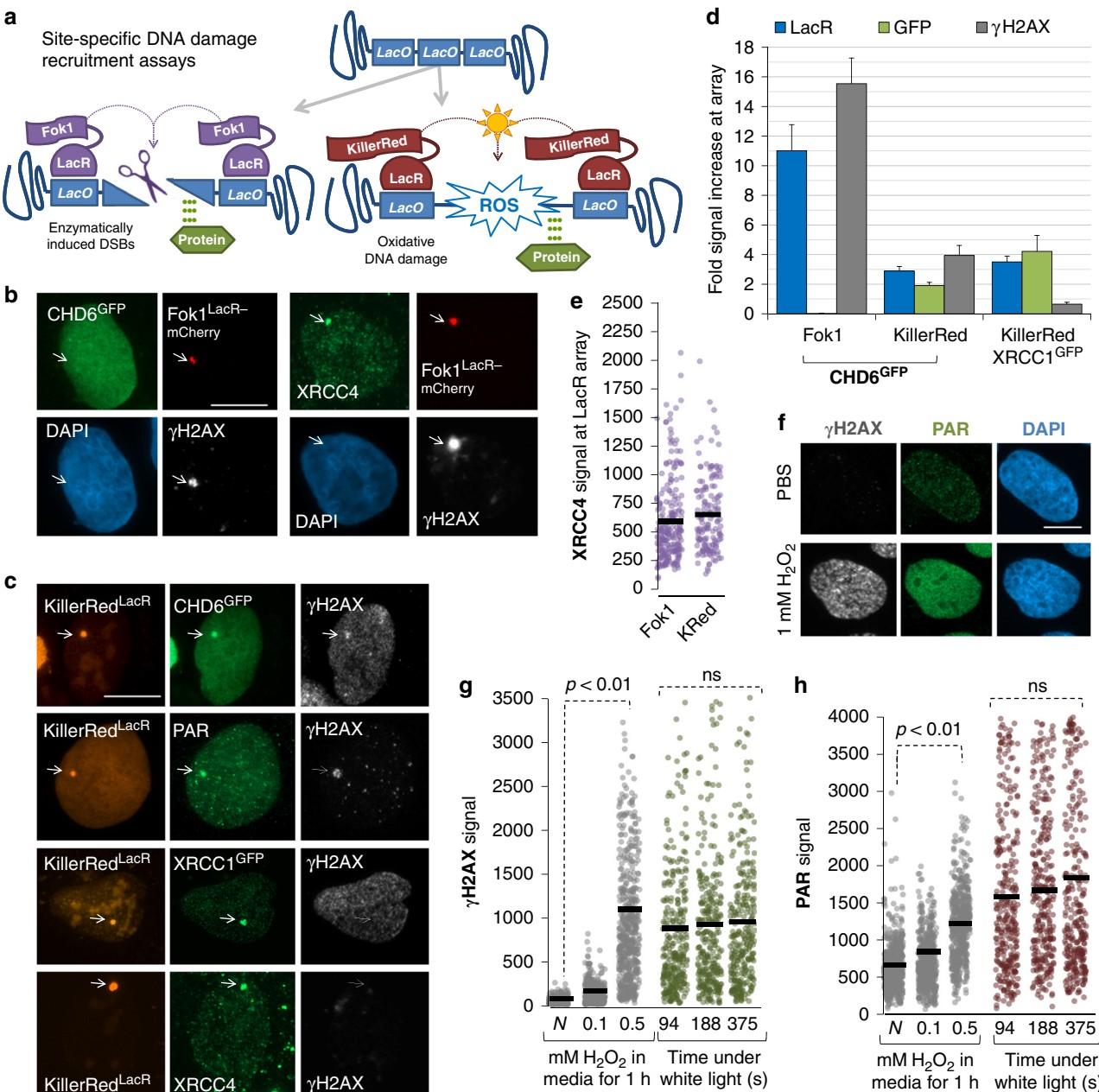

**Fig. 2** CHD6 is recruited selectively to oxidative lesions. **a** Schematic of site-specific DNA damage recruitment assay. **b**, **c** Fok1 or KillerRed expression was induced in U2OS 2-6-3 cells (transiently expressing CHD6^GFP, XRCC1^GFP or neither) then immunostained with γH2AX, PAR and/or XRCC4 and imaged. **d**, **e** The fold increase in γH2AX, LacR, XRCC4, XRCC1 and/or GFP signal in cells from (**b**, **c**) was quantified. Error bars = s.e.m.; $n = 3$. **e**–**h** U2OS 2-6-3 cells were exposed to 1 mM $H_2O_2$ in media for 1 h, immunostained and imaged for nuclear γH2AX and PAR signal and compared to KillerRed expressing cells stimulated with 94–375 s of white light. $P$ values are for Student's $t$-test. All scale bars equal 5 μm

in cells treated with PARG inhibitors (PARGi) or PARG siRNA, suggesting that CHD6 binds to PAR directly or to PAR-binding proteins (Fig. 3a–c). XRCC1 is a major PAR-binding platform for recruiting DNA damage response proteins[28,29] and accumulated fourfold more than CHD6 at micro-irradiation tracks with dissimilar recruitment and dispersal kinetics (Fig. 1i). XRCC1 loss by CRISPR/Cas9 (clustered regularly interspaced short palindromic repeats/CRISPR-associated 9)-mediated deletion did not impact CHD6^GFP recruitment, suggesting that CHD6 binds to PAR directly or indirectly via an XRCC1-independent process (Supplementary Fig. 1g).

To interrogate the specific regions of CHD6 required for recruitment to DNA damage sites, we introduced multiple point

or truncation mutations without perturbing nuclear localization (Fig. 3d, Supplementary Fig. 1h). Ablation of a conserved catalytic amino acid (K492A) had no impact on CHD6 dynamics (Fig. 3e). Introduction of point mutations into highly conserved amino acids (F318A, Y322A, Y398A, W402A=ΔCD1+2) known to ablate CHD enzyme ability to bind methyl-lysine[30,31] perturbed mid-to-late retention at DNA damage, although it had no effect on the early/fast recruitment (Fig. 3e). Deletion of the C-terminal BRK domain (1–1448) had no effect on CHD6 dynamics, while additional loss of the SANT domain (1–1028) and/or the ATPase/Helicase core (1–449) only impacted late retention (Fig. 3e). Further loss of the double chromodomain ablated the later accumulation and retention of CHD6 (1–269), suggesting (similar

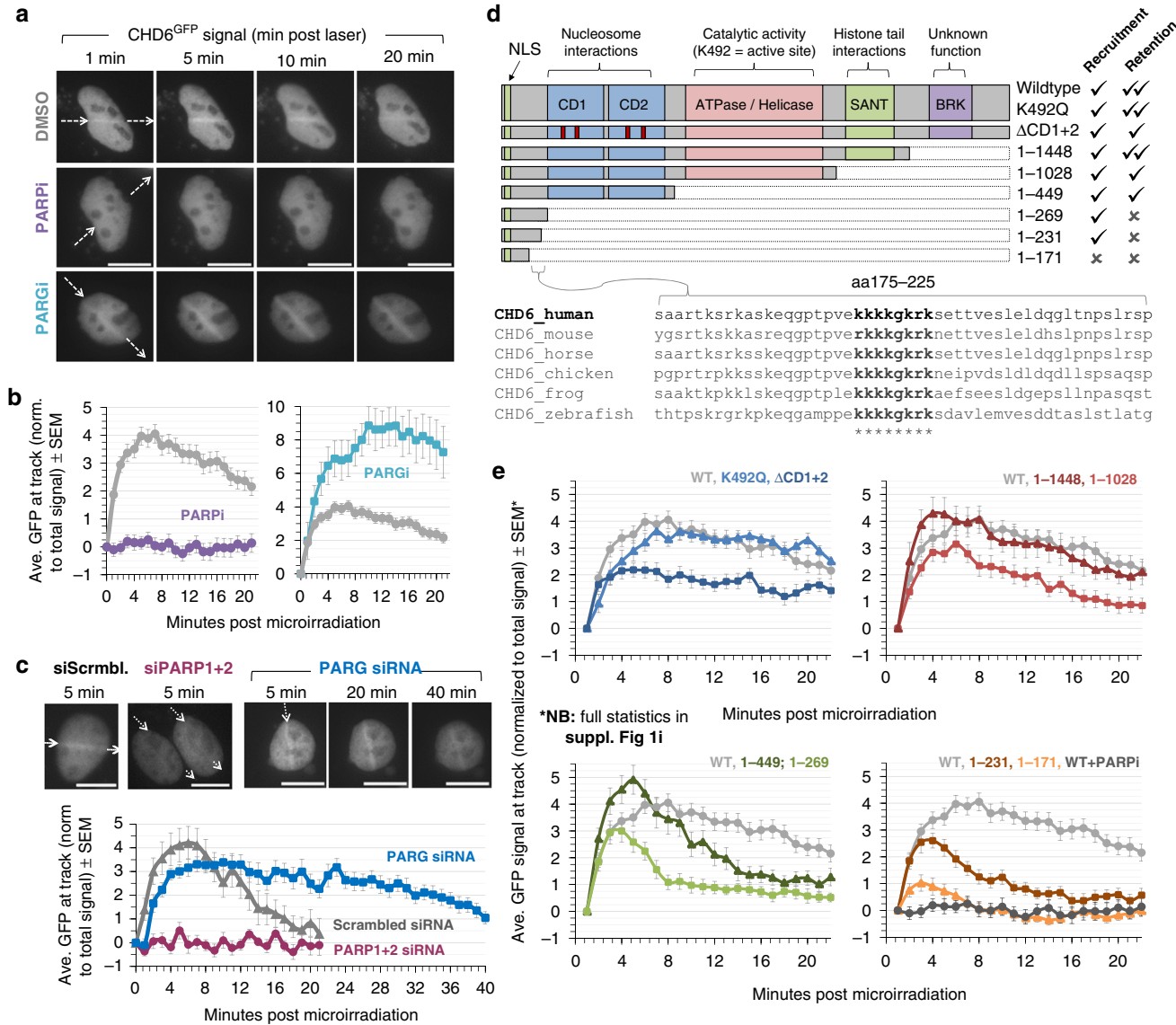

**Fig. 3** CHD6 recruitment to DNA damage is PAR-dependent. **a** A549 expressing CHD6[GFP] were treated with DMSO, PARP inhibitor (PARPi) or PARG inhibitor (PARGi) for 1 h before micro-irradiation and live imaging over 21 min. Scale bars equal 10 μm. **b** Cells were quantified for GFP signal increase within the laser track as described in Methods. Gray lines indicate DMSO-treated cells; purple lines indicate PARPi-treated cells; cyan lines indicate PARGi-treated cells. Error bars = s.e.m.; n = 8–30. **c** A549 were transfected with indicated siRNAs and CHD6[GFP]; 16 h later, cells were microirradiated and live imaged and quantified as in (**a**, **b**). Scale bars equal 10 μm. Gray lines indicate scrambled siRNA-treated cells; magenta lines indicate PARP1 and PARP2 siRNA-treated cells; blue lines indicate PARG siRNA-treated cells. Error bars = s.e.m.; n = 20. **d** Schematic of known CHD6 domains, including: catalytic dead (K492Q), double chromodomain (ΔCD1+2) and truncation mutations. Cross-species alignment of the PAR-dependent recruitment region is expanded. **e** A549 expressing CHD6 mutants indicated in (**d**) were treated and analyzed as in (**a**, **b**). All error bars = s.e.m.; n = 12–53. Full statistical analysis for data in all panels is in Supplementary Table 1. Line colors are as follows: gray = WT; light blue = K492Q; dark blue = ΔCD1+2; dark red = 1–1448; light red = 1–1028; dark green = 1–449; light green = 1–269; dark orange = 1–231; light orange = 1–171; dark gray = WT treated with PARPi

to the ΔCD1+2 mutant) that the double chromodomain is required for the second phase of CHD6 relocalization to sites of DNA damage, after initial PAR-dependent recruitment (Fig. 3e). Further deletion analysis revealed that the early, fast, PAR-dependent recruitment of CHD6 was attributable to aa171–231, with the 1–171 mutant unable to relocalize to DNA damage (Fig. 3e) and statistically similar to PARPi-treated cells expressing wild-type CHD6 (Supplementary Table 1). This 60-amino acid (aa) region contains a single, highly conserved, positive KR-rich patch similar to regions that enable electrostatic interactions with PAR in other proteins[32] (Fig. 3d). Collectively, these data show that CHD6 is recruited to DNA damage sites in a manner dependent initially on its N-terminal PAR-dependent binding

region, and then retained subsequently via its double chromodomain and SANT domain.

To corroborate our micro-irradiation data with an alternative approach, we next attempted to examine whether CHD6 interacts with endogenous PAR by pulling down GFP or CHD6[GFP]. We used HEK293 cells, as A549 displayed insufficient transfection efficiency. Cells were treated with PARGi (to block PAR degradation) and $H_2O_2$ to stably induce PAR; however, we found that PAR was strongly suppressed by CHD6[GFP] over-expression, precluding the experiment (Supplementary Fig. 1i). We speculated that this might represent suppressed ROS-induced DNA damage (and hence PAR) when CHD6 is in abundance. Alternatively, we expressed GFP or GFP-tagged CHD6 or

CHD4 (as a positive control) and incubated extracts with streptavidin-agarose loaded with biotin alone, PAR-biotin or $H3^{K9me3}$-biotin (a canonical CHD-binding epitope), and immunoblotted for GFP. GFP-tagged CHD6 and CHD4 (but not GFP alone) were present in PAR- or $H3^{K9me3}$-biotin pulldowns, but not biotin alone (Supplementary Fig. 1j). This confirmed our observations using lasers, demonstrating that CHD6 displays PAR-dependent binding activity.

**CHD6 ablation leads to elevated PAR during oxidative stress.** To assess the significance of CHD6 expression under resting and oxidative stress conditions, we perturbed *CHD6* in A549 cells using CRISPR/Cas9-mediated gene editing (ΔCHD6). This ablated endogenous CHD6 by immunofluorescence or immunoblot, was comparatively superior to siRNA-based depletion (in terms of endogenous CHD6 signal loss) and also validated CHD6 antibody specificity (Fig. 4a). Single cell cloned A549$^{ΔCHD6}$ lines were analyzed by multiple independent gene sequencing runs, indicating a 100 nucleotide deletion from both *CHD6* alleles. PCR amplification of the guide RNA target region confirmed a deletion within CHD6 exon 2 (Supplementary Fig. 2a), while immunofluorescence confirmed protein loss and our ability to re-express CHD6$^{GFP}$ ectopically (Fig. 4a, Supplementary Fig. 2b). A549$^{ΔCHD6}$ were viable, grew well under unstressed conditions and, by fluorescence-activated cell sorting (FACS) analysis, showed no abnormalities in the cell cycle phase distribution of asynchronous populations (Supplementary Fig. 2c, d). The expression of PARP1 and its relocalization to DNA damage was unaltered in A549$^{ΔCHD6}$ (Supplementary Figs. 2e, f). The CHD1, CHD2, CHD3.1 and CHD4 expression also remained unchanged (Supplementary Fig. 2g). We next ascertained whether PAR induction was indeed altered by CHD6 expression status, as our earlier result implied (Supplementary Fig. 1i). We were unable to generate lines stably overexpressing CHD6, due to repeatedly observed silencing of *CHD6* constructs during selection. Hence, we relied upon transient transfection to rescue A549$^{ΔCHD6}$ phenotypes. As before, PARGi was used to stabilize PAR. PAR induction by $H_2O_2$ was ~2-fold greater in A549$^{ΔCHD6}$ compared to controls by immunoblot or immunofluorescence, and this was suppressed by CHD6$^{GFP}$, but not GFP or CHD4$^{GFP}$ overexpression (Fig. 4b–d, Supplementary Fig. 3a, b), indicating the genetic specificity of our system. Similar effects were observed in the absence of PARGi, albeit with much weaker signal (Supplementary Fig. 3c). CHD6$^{GFP}$ over-expression (but not GFP or CHD4$^{GFP}$) also suppressed PAR in wild-type cells (Fig. 4b–d, Supplementary Fig. 1i), fitting with earlier results. CHD6$^{ΔCD1+2}$ failed to complement A549$^{ΔCHD6}$, suggesting that retention at sites of oxidative DNA damage is functionally important. CHD6 mutated in the ATPase/helicase domain (K492Q) also failed to complement A549$^{ΔCHD6}$, suggesting that CHD6 catalytic activity is necessary for its role in regulating the oxidative DNA damage response (Fig. 4d). These data indicate a role for CHD6 in controlling PAR induction following oxidative stress-induced DNA damage, an activity that involves both its ability to remodel chromatin and to be retained at sites of oxidative DNA damage.

**Cells lacking CHD6 have diminished antioxidant responses.** PAR formation is triggered by oxidative DNA damage and to directly measure the intrinsic cellular oxidative stress of CHD6-deleted cells, we incubated cells with chemically reduced (non-fluorescent) 2',7'-dichlorodihydrofluorescein diacetate (H2DCFDA) which, upon oxidation, converts to fluorescent 2',7'-dichlorofluorescein (DCF) and measured fluorescence±$H_2O_2$ by FACS (Fig. 4e). In the resting state, A549$^{ΔCHD6}$ populations contained 1.65-fold more DCF-positive cells relative to wild type

and 2.9-fold more following $H_2O_2$, confirming a state of elevated cellular oxidative stress in the absence of CHD6 (Fig. 4e). Wild-type CHD6$^{mPLUM}$ (GFP could not be used, as it interfered with green DCF signal) complemented the elevated oxidative stress observed in A549$^{ΔCHD6}$ cells treated with $H_2O_2$, further confirming specificity (Fig. 4e). Little is known about the detailed molecular consequences of CHD6 deletion, although there is one report of a *yeast-two-hybrid* interaction between CHD6 and Nrf2[33], which is important in controlling antioxidant transcriptional events[34]. To explore this using our validated human genetic model of CHD6 deletion, we used qPCR to measure Nrf2 gene target expression after chronic $H_2O_2$ over 24 h. This included genes highly responsive to $H_2O_2$: *HMOX1* (heme oxygenase 1), a regulator of iron homeostasis and Fenton reactions producing OH radicals from $H_2O_2$, and *TXNRD1* (thioredoxin reductase 1) that regenerates thioredoxin following reduction of proteins oxidized by OH radicals. While *HMOX1* and *TXNRD1* gene expression both increased within 4-8 h of $H_2O_2$ in wild-type cells, there was suppressed baseline expression and comparatively low-to-no increased expression after $H_2O_2$ in A549$^{ΔCHD6}$ (Fig. 4f). We also examined the expression of two other Nrf2 targets: *NQO1* (NADPH:Quinine Oxidoreductase 1) involved in o-quinone detoxification and *G6PD* (glucose-6-phosphate dehydrogenase) involved in NADPH synthesis. There was no $H_2O_2$-inducible increase in *NQO1* or *G6PD* expression, indicating these genes are not upregulated by $H_2O_2$ in these cells, with minor to insignificant differences in A549$^{ΔCHD6}$ (Fig. 4f, g, Supplementary Fig. 3d). To control for confounding effects on transcription in general, we examined *TBP*, a gene unrelated to oxidative stress; there was no change in *TBP* expression with $H_2O_2$ or in A549$^{ΔCHD6}$ versus wild type (Fig. 4g). PARP inhibition had no impact on $H_2O_2$-induced *HMOX1* or *TXNRD1* induction, indicating that this role of CHD6 is likely distinct to its PAR-dependent DNA damage recruitment (Supplementary Fig. 4a). To assess the impact of antioxidant gene expression on the A549$^{ΔCHD6}$ phenotype, we transiently over-expressed GFP-tagged HMOX1 and assessed PAR induction after $H_2O_2$ as in Fig. 4d. At even low to moderate levels of expression, HMOX1$^{GFP}$ (but not GFP alone) suppressed the elevated PAR induction phenotype of A549$^{ΔCHD6}$ (Fig. 5a). These data are indicative of a role for CHD6 in the antioxidant transcriptional response involving a sub-pathway of Nrf2-controlled gene expression, and may partly explain how CHD6 controls PAR formation after oxidative stress.

**CHD6 loss elevates DNA damage with no overt DNA repair defect.** Elevated oxidative stress, reduced anti-oxidants and exaggerated PAR induction are all suggestive of increased DNA breakage that can be monitored by comet assays and immunofluorescence of γH2AX or 53BP1 foci[35]. Alkaline comet assays to measure SSB induction and repair revealed an increase in SSBs formed after $H_2O_2$ in A549$^{ΔCHD6}$ relative to controls (Fig. 5b, c), albeit there was no significant change in actual SSB repair (Fig. 5d). 53BP1 foci per cell, with a wide distribution within a population, increased with acute $H_2O_2$ exposure and were also markedly higher in A549$^{ΔCHD6}$ (Fig. 5e). Looking at foci per cell over time, 53BP1 foci resolution was not impacted by CHD6 loss, consistent with a role in suppressing DNA damage induction, rather than DSB repair itself. A similar, although much more subtle, phenotype was observed after IR exposure (Fig. 5f). This further contrasts the role of CHD6 within the DNA damage response from CHD1–CHD4, the loss of any of which impacts DSB repair capacity[3–13]. To more comprehensively determine the DNA repair capacity in the absence of CHD6, we used fluorescence multiplex host cell reactivation (FM-HCR) assays to

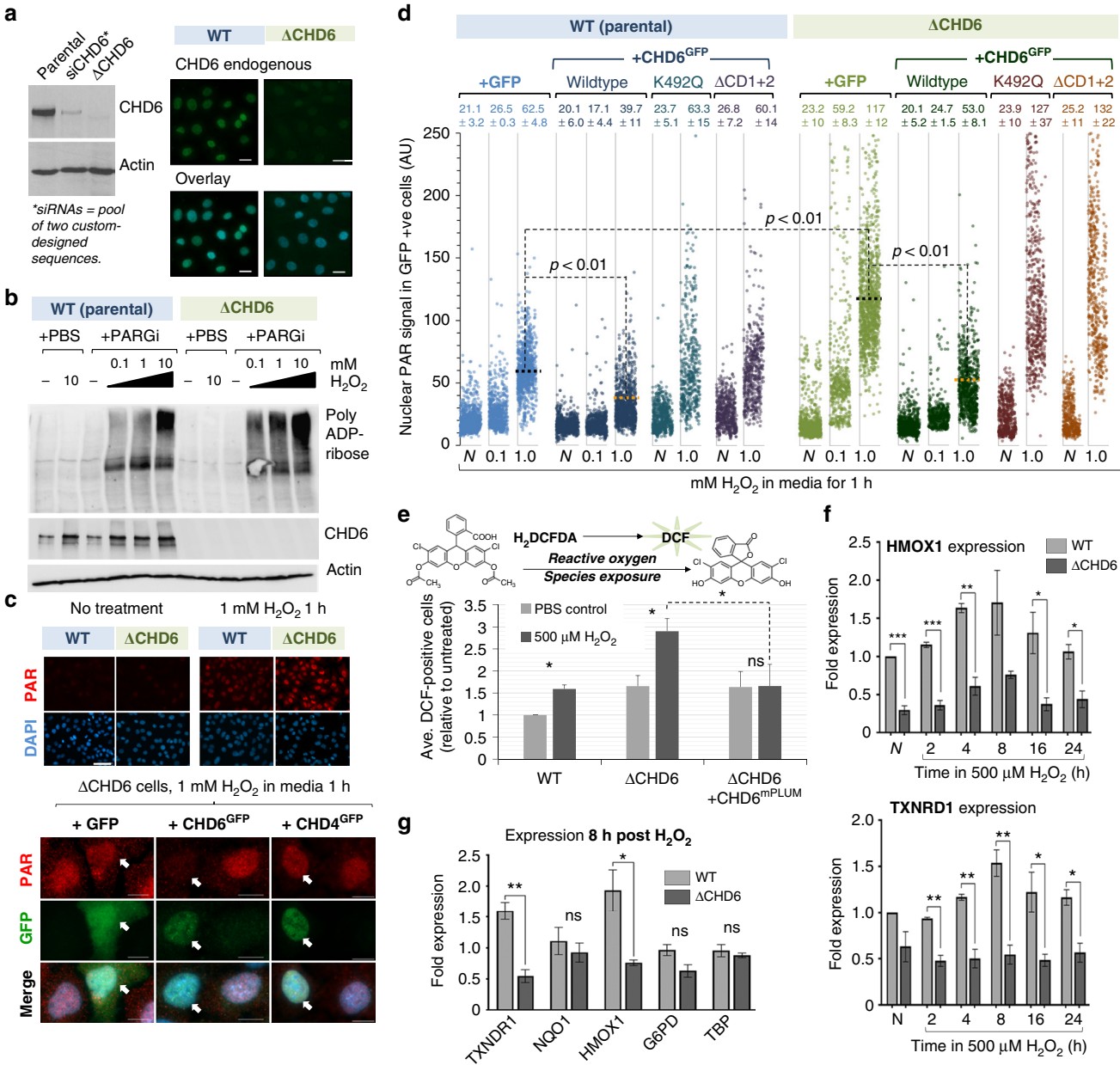

**Fig. 4** Cells lacking CHD6 display elevated PAR, oxidative stress and have diminished antioxidant responses. **a** A549 were subjected to CRISPR-mediated CHD6 gene ablation (ΔCHD6) or treated with CHD6 siRNA (siCHD6) before being immunoblotted (left) or immunostained (right) as indicated. Parental refers to the wild-type (WT) A549 used to derive ΔCHD6. Scale bars = 20 μm. **b** WT and A549ΔCHD6 were treated ±5 μM PARGi and indicated $H_2O_2$ doses for 1 h before immunoblotting for PAR, CHD6 and actin. Representative blot shown, $n = 3$. **c** WT and ΔCHD6 A549 were complemented with GFP, CHD6GFP or CHD4GFP and treated with 5 μM PARGi ± 1 mM $H_2O_2$ in media for 1 h before immunostaining for PAR (red) and DAPI (blue). Lower panels highlight GFP-expressing cells. Scale bars = 10 μm. **d** WT and A549ΔCHD6 were transfected with GFP, or GFP-tagged CHD6wildtype, CHD6Δchromo or CHD6K492Q before treatment with 5 μM PARGi ± 0–1 mM $H_2O_2$ for 1 h and immunostaining for PAR and GFP as in (**c**). Nuclear PAR in >1000 GFP-positive cells ($n = 3$) was quantified using ImageJ. Dot colors represent: light blue = GFP alone, WT cells; dark blue = WT CHD6GFP, WT cells; cyan = K492Q CHD6GFP, WT cells; purple = ΔCD1+2 CHD6GFP, WT cells; light green = GFP alone, ΔCHD6 cells; dark green = WT CHD6GFP, ΔCHD6 cells; dark red = K492Q CHD6GFP, ΔCHD6 cells; orange = ΔCD1+2 CHD6GFP, ΔCHD6 cells. Error bars = s.e.m.; $n = 3$. **e** WT, A549ΔCHD6 and A549ΔCHD6 complimented with CHD6mPLUM, incubated with H2DCFA then exposed ± 0.5 mM $H_2O_2$ for 1 h and FACS analysis of green fluorescent DCF signal. Data represent an average of the percent DCF-positive cells within a population, expressed relative to the PBS-treated wild-type control. Error bars = s.e.m.; $n > 3$. **f, g** WT and A549ΔCHD6 were exposed to ±500 μM $H_2O_2$ in media for up to 1 day and analyzed by qPCR to ascertain HMOX1, TXNRD1, NQO1, G6PD and TBP mRNA expression. Error bars = s.e.m.; $n = 3$. In all cases, $P$ values are for Student's $t$-test: ns not significant ($P > 0.05$); *$P < 0.05$; **$P < 0.01$; ***$P < 0.001$. In all cases, ± refers to with and without

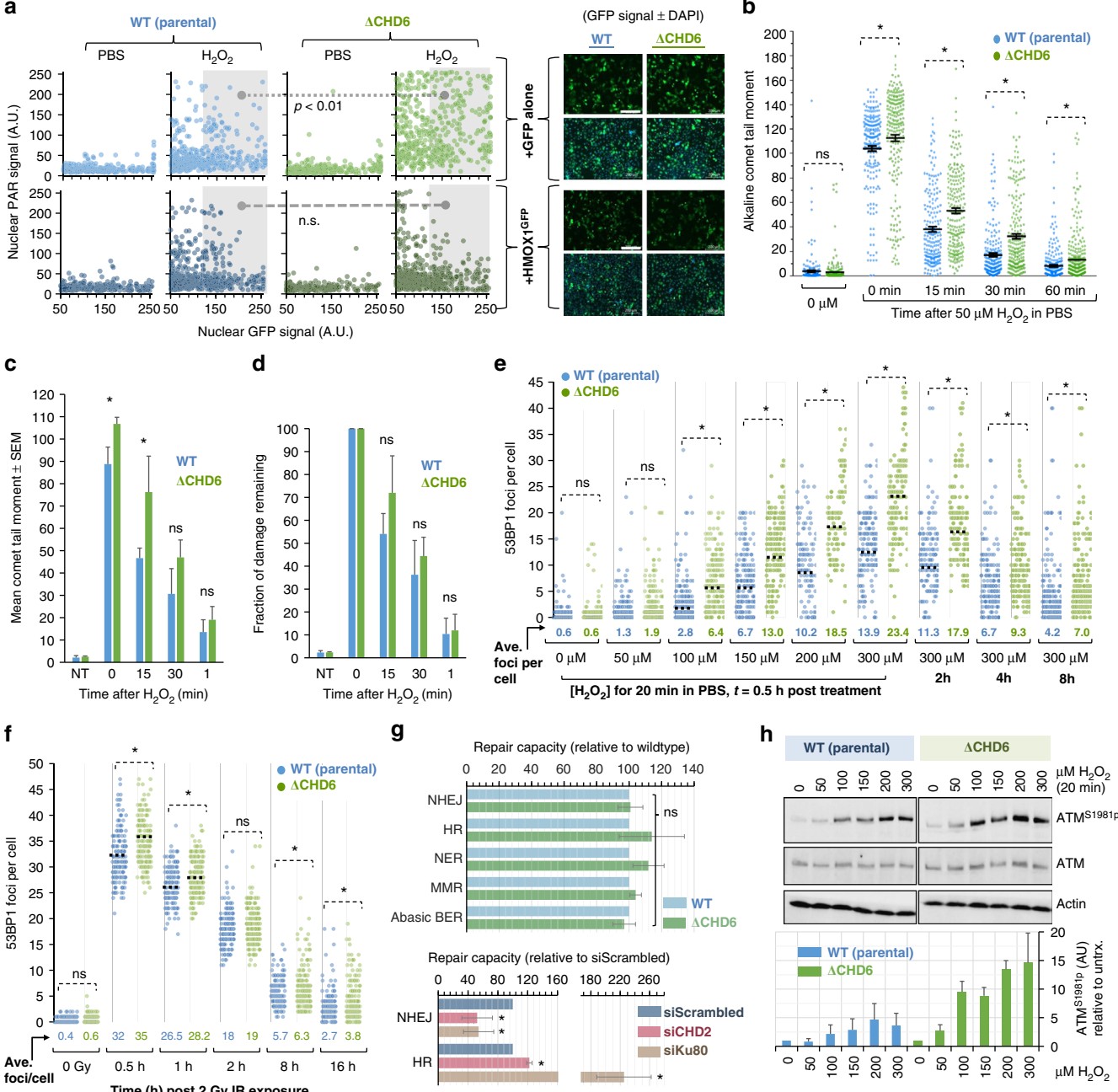

**Fig. 5** CHD6-deficient cells display increased single-strand breakage, DNA damage signaling but normal DNA repair capacity. **a** WT and ΔCHD6 A549 were transfected with GFP or HMOX1^GFP and treated with 5 μM PARGi ± 1 mM H₂O₂ in media for 1 h before immunostaining for PAR. Nuclear PAR and GFP signal was quantified by ImageJ. Right panels = representative images indicating GFP (green) and/or DAPI (blue) signal in each condition. **b–d** WT and ΔCHD6 A549 in suspension were treated with 50 μM H₂O₂ and analyzed by alkaline comet assay as in Methods; **b** raw data from single experiment; **c** mean comet tail moment for $n = 3$; error bars = s.e.m.; **d** data normalized to induced damage (0 min time point). Error bars = s.e.m. **e** WT and A549^ΔCHD6 were exposed to 0–300 μM H₂O₂ in PBS for 20 min, washed, returned to media for 0.5–8 h, fixed and immunostained for 53BP1. 53BP1 foci per cell were enumerated for >50–100 cells per $n$, $n = 3$. Graph indicates scatters of foci per cell across population; average number of foci per cell are indicated. **f** Cells from (**a**) were irradiated and harvested at indicated time points, then processed and analyzed as in (**a**). **g** (Top) WT, A549^ΔCHD6 were incubated for 4 h at 3% O₂ for acclimatization and transfected with FM-HCR reporter plasmids (see Methods). A549^ΔCHD6 repair capacity for each substrate was normalized to WT values, and is represented as relative repair capacity. (Bottom) A549 were transfected (48 h earlier) with scrambled, CHD2 or Ku80 siRNA, and transfected 48 h later with FM-HCR plasmids. Error bars = s.e.m.; $n = 3$. **h** WT and A549^ΔCHD6 were exposed to 0–300 μM H₂O₂ in PBS for 20 min, washed, returned to media for 0.5 h and immunoblotted for ATM^S1981p, ATM and actin. Average signal from $n = 3$ is indicated in lower panel, error bars = s.d. In all cases, $P$ values are for Student's $t$-test: ns not significant ($P > 0.05$); *$P < 0.05$. In all cases, ±refers to with and without, blue refers to WT cells; green refers to ΔCHD6 cells

measure: (i) non-homologous end-joining (NHEJ)-mediated DSB repair, (ii) homologous recombination (HR)-mediated DSB repair, (iii) nucleotide excision repair (NER) of ultraviolet C-induced DNA lesions, (iv) mismatch repair (MMR) of a G:G mispair and (v) BER of a tetrahydrofuran abasic site analog. In brief, fluorescent reporter plasmid substrates containing controlled quantities of each specific lesion are transfected into cells, whereupon DNA repair alters fluorescent reporter signal detectable by flow cytometry; these assays have been validated previously[36–38]. CHD6 deletion did not significantly impact NHEJ or HR-mediated DSB repair or BER pathways requiring SSB repair (Fig. 5g), fitting with 53BP1 foci kinetic and comet assay data. As controls, we depleted either CHD2 or Ku80 by siRNA and observed an expected[3] reduction in NHEJ and/or increase in HR. Ku80 knockdown confirms the ability of FM-HCR to detect siRNA-induced DSB repair defects, while CHD2 knockdown indicates that FM-HCR is sensitive to altered CHD status at least for NHEJ and HR endpoints (Fig. 5g). No defects in MMR, NER or BER-mediated DNA repair were observed in A549$^{\Delta CHD6}$ relative to wild type using FM-HCR, and although this will require future work to confirm, it suggests that CHD6 may not play a role in these DNA repair pathways.

**Cells lacking CHD6 display amplified DSB signaling.** Fitting with increased induction of oxidative DNA damage, A549$^{\Delta CHD6}$ displayed greater ATM$^{S1981p}$ signal (a marker of DSB-induced ATM protein kinase signal activation) for a given dose of $H_2O_2$ compared to controls (Fig. 5h). This was further supported by a cell-by-cell analysis of H2AX$^{S139p}$ and KAP1$^{S824p}$ signal, both downstream readouts of ATM signaling activity[39] and, for a range of $H_2O_2$ doses and time points, the signal intensity of H2AX$^{S139p}$ and KAP1$^{S824p}$ were both greater in A549$^{\Delta CHD6}$ versus controls (Fig. 6a). To explore whether this increase in DSB response signal transduction impacted cell cycle checkpoint arrest, a primary functional endpoint of the DSB response, we monitored H3$^{S10p}$ and nuclear morphology as a read out of mitotic entry, distinguishing G2 from M phase in asynchronous populations[40]. Complete G2/M checkpoint arrest was observed in A549$^{\Delta CHD6}$ at $H_2O_2$ doses too low to trigger complete arrest in controls (Fig. 6b). At doses and times with the largest differences in G2/M arrest, only minor differences in 53BP1 foci/cell were seen, suggesting checkpoint hypersensitivity in A549$^{\Delta CHD6}$ cannot be explained entirely by DSB numbers. To explore this further, we repeated this experiment using IR (Fig. 6b), observing a similar hypersensitive arrest phenotype with little difference in 53BP1 foci and, by re-plotting data as mitotic index relative to foci number/cell, we determined that A549$^{\Delta CHD6}$ showed a 2–3-fold increase in G2/M checkpoint sensitivity regardless of DSB number or source (Supplementary Fig. 4b). These findings suggest that CHD6 may have a pleiotropic impact on oxidative DNA damage responses, potentially involving suppression of damage induction (via antioxidant upregulation) and/or modulating ATM signaling to define cell cycle checkpoint arrest sensitivities.

**CHD6 regulates chromatin compaction distinct from CHD3.** We hypothesized that a disrupted chromatin state may contribute to amplified DNA damage signaling of CHD6-ablated cells—in essence, a multiplying effect alongside increased oxidative damage induction. The sensitivities of DNA damage response endpoints (such as cell cycle checkpoint arrest) are generated by the signal transduction mechanisms of the cell phase in question; for example, the G2/M checkpoint requires ATM signaling produced by 10 or more DSBs to remain 100% active, below which signaling is insufficient to withhold cells from mitosis[40]. Cells with abnormally relaxed chromatin are known to display hypersensitive G2/M checkpoint activation, as ATM signaling per DSB is increased in the absence of the dampening effect of nucleosome compaction[41]. Comprehensive mass spectrometry analysis of the epigenetic profile of A549$^{\Delta CHD6}$ relative to wild type revealed no significant changes in histone H3 or H4 acetylation or methylation marks linked to either open or compacted chromatin (Fig. 6c). However, when we assessed A549$^{\Delta CHD6}$ chromatin sensitivity to limited micrococcal nuclease (MNase) digestion, which cleaves nucleosome linker DNA and is more active in relaxed chromatin[9], A549$^{\Delta CHD6}$ displayed increased MNase sensitivity (Fig. 6d), indicative of CHD6 promoting compaction at, overall, more genomic regions than it relaxes and providing a plausible explanation for the exaggerated ATM signaling and G2/M checkpoint sensitivity in these cells.

The increased MNase sensitivity and greater DNA damage signaling in A549$^{\Delta CHD6}$ is reminiscent of effects caused by perturbed histone H1 occupancy[42,43] or depleting the CHD3 or KAP1 constitutive heterochromatin builders[9,10,41,44]. Histone H1 contributes to chromatin relaxation by being displaced via PAR and re-incorporated by an uncertain mechanism[42,43,45]. We found no difference in H1 displacement or reincorporation dynamics in A549$^{\Delta CHD6}$ cells compared to wild type (Supplementary Fig. 4c). Immunofluorescence analysis of euchromatin markers H3$^{K4me3}$ or H4$^{K8ac}$ or constitutive heterochromatin markers H3$^{K9me3}$ and KAP1 confirmed no gross alterations in A549$^{\Delta CHD6}$ (Supplementary Fig. 4e). We further examined this by assessing the ATM dependency of DSB repair, as cells with perturbed constitutive heterochromatin (e.g., via CHD3 depletion) show no need for ATM activity during repair (Fig. 6e)[9,10,44]. A549$^{\Delta CHD6}$ cells displayed normal ATM dependency, suggesting that CHD3-dependent heterochromatin is functionally independent of CHD6 status, at least in the context of the DNA damage response (Fig. 6e). Altogether, these data raise the possibility that CHD6 regulates chromatin compaction or accessibility in such a way that is apparently distinct to CHD3, but elicits a comparable increase in ATM signaling when absent[9,10,36,39].

**CHD6 loss leads to failure to thrive after oxidative stress.** Finally, we performed survival and proliferation experiments after acute ($H_2O_2$ in PBS for 20 min, then removed) or chronic ($H_2O_2$ added to media, refreshed daily) oxidative stress. Relative to controls, the clonogenic potential of A549$^{\Delta CHD6}$ 10 days after acute $H_2O_2$ exposure trended downwards, although this lacked statistical significance (Fig. 7a). This result was clarified by cell growth analysis, which revealed that although A549$^{\Delta CHD6}$ cells undergo a significant attrition within 48 h of acute $H_2O_2$, they repopulate within 8 days (Fig. 7b). In contrast, wild-type controls only slowed growth after acute $H_2O_2$ and recover completely within 8 days (Fig. 7b). After chronic oxidative stress exposure, A549$^{\Delta CHD6}$ clonogenic survival was reduced by 1–3 orders of magnitude in comparison to controls (Fig. 7c). Similarly, while wild-type cells were unaffected by chronic exposure to 5–50 μM $H_2O_2$ and only experience growth stasis at 500 μM $H_2O_2$, the A549$^{\Delta CHD6}$ attenuated growth in 5 μM $H_2O_2$ and exhibited significant dose-dependent cell death in 50–500 μM $H_2O_2$ (Fig. 7d). Identical effects were seen when cells were grown at either 21% or 3% oxygen (Fig. 7e vs. Supplementary Fig. 4d). We attempted to restore CHD6 expression and monitor cell growth; however, these experiments were precluded by cell death observed in CHD6-overexpressing cells >4 days after transfection. We speculated this was due to the acrimonious long-term impact of an overexpressed chromatin remodeling enzyme potentially lacking stoichiometric levels of as yet unknown regulatory factors. In lieu, we expressed HMOX1 to address to what extent CHD6-dependent antioxidant responses contribute to cell survival

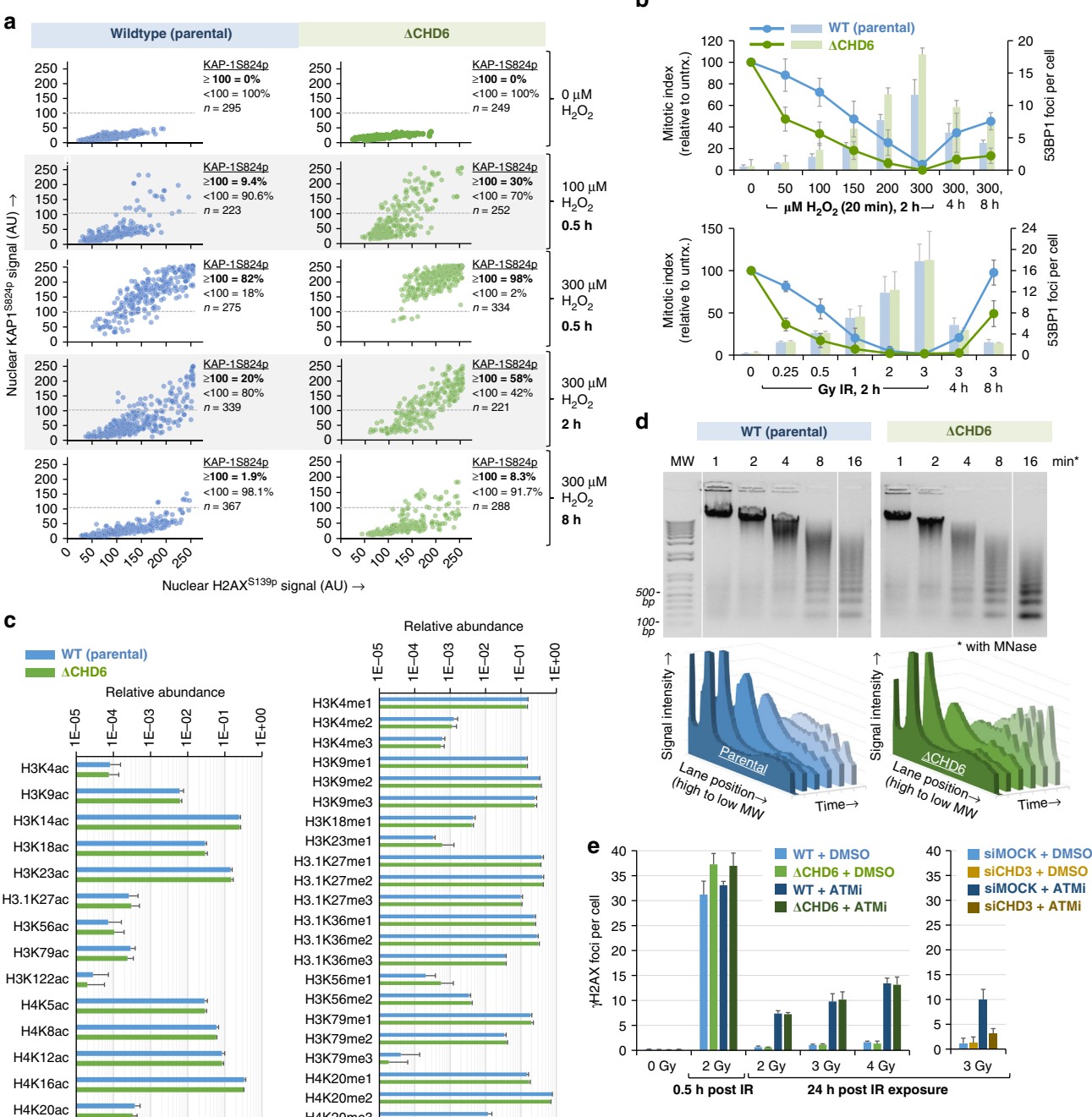

**Fig. 6** CHD6-deficient cells have hypersensitive cell cycle checkpoint arrest, abnormally relaxed chromatin but functionally normal constitutive heterochromatin and epigenetics. **a** WT and A549$^{\Delta CHD6}$ were exposed to 0–300 µM $H_2O_2$ in PBS for 20 min, washed, returned to media for 0.5–8 h, immunostained and quantified for nuclear KAP1$^{S824p}$ and H2AX$^{S139p}$ signal (200–300 cells across $n$ of 3). Percentage cells with $\geq$ or <100 KAP1$^{S824p}$ signal (demarcating substantial DNA damage signaling) is indicated. Data represent $n = 3$. **b** WT and A549$^{\Delta CHD6}$ were exposed either to 0–300 µM $H_2O_2$ in PBS for 20 min, washed, returned to media for 2–8 h, or irradiated with 2 Gy IR before being fixed and immunostained for 53BP1 (demarcating DSBs), DAPI and H3$^{S10p}$ (to demarcate G2/M phase cells). Mitotic indexes (lines, left $y$-axis) were derived from the % M-phase cells (=strongly positive H3$^{S10p}$ signal and DAPI morphology) relative to control. The 53BP1 foci per cell (bars, right $y$-axis) were also enumerated. Error bars = s.e.m.; $n = 3$. **c** 1 × 10$^7$ WT and A549$^{\Delta CHD6}$ were processed via acid extraction to isolate histones, which were then derivatized with propionic anhydride, digested and quantified for the indicated epigenetic modifications by mass spectrometry (see Methods). Error bars = s.d.; $n = 6$. **d** Nuclei isolated from confluent WT and A549$^{\Delta CHD6}$ were exposed to MNase for indicated times before isolation and resolution of genomic DNA by agarose gel. Representative ethidium bromide-stained 1.2% agarose gels are shown (upper panels, inverted signal) alongside quantified DNA signal across each lane (lower panels) from a total of $n = 3$. **e** WT and A549$^{\Delta CHD6}$ were transfected with scrambled, CHD6 or CHD3 siRNA 48 h before incubation with DMSO or 10 µM ATMi 30 min prior to irradiation, fixation at indicated times, γH2AX immunostaining and foci enumeration. Error bars = s.e.m.; $n = 3$. In all cases, blue refers to WT cells; green refers to ΔCHD6 cells and yellow refers to WT cells treated with CHD3 siRNA

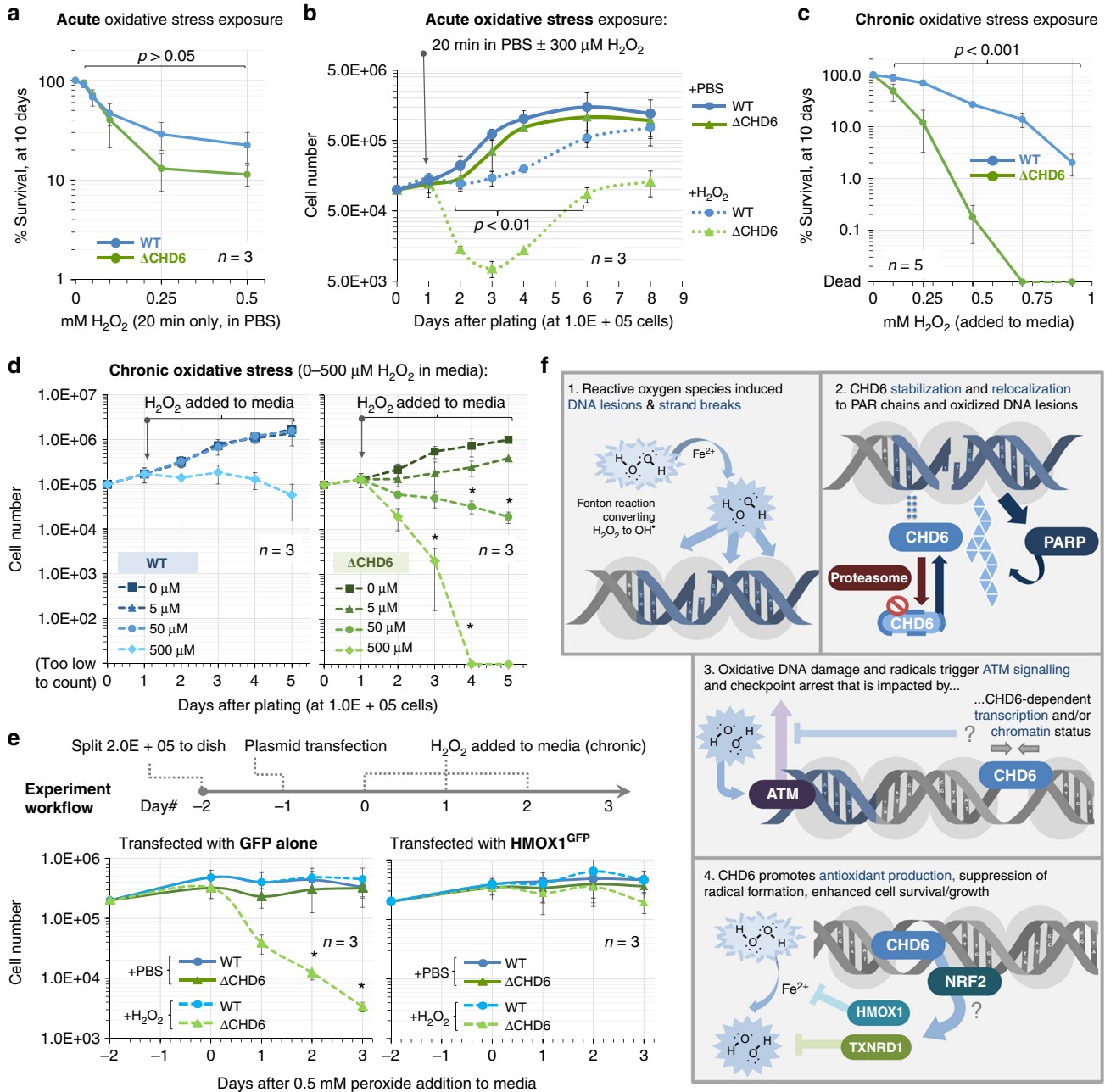

**Fig. 7** CHD6 loss causes failure to thrive after oxidative stress and perturbed oxidative base excision repair initiation. **a** WT and A549$^{\Delta CHD6}$ cells were exposed to $H_2O_2$ in PBS for 20 min, washed, plated and scored for colony formation 10 days later. Error bars = s.e.m.; $n = 3$; $P$ value determined by two-way ANOVA. **b** The 1.0E + 05 WT and A549$^{\Delta CHD6}$ were plated and, 24 h later, exposed to ±300 µM $H_2O_2$ in PBS for 20 min, washed and returned to media for 8 days. Viable cells were counted daily. Error bars = s.e.m.; $n = 3$. $P$ values represent unpaired $t$-test between WT and A549$^{\Delta CHD6}$ at indicated time points. **c** The 300× WT and A549$^{\Delta CHD6}$ cells were exposed to $H_2O_2$ in media for 48 h, plated and scored for colony formation 10 days later. Error bars = s.e.m.; $n = 5$; $P$ value determined by two-way ANOVA. **d** The 1.0E + 05 WT and A549$^{\Delta CHD6}$ were plated and, 24 h later, 0–500 µM $H_2O_2$ was added to media. Fresh $H_2O_2$ was added daily to maintain chronic exposure. Cells were analyzed as in (**b**). Error bars = s.e.m.; $n = 3$. **e** WT and A549$^{\Delta CHD6}$ were seeded at 2.0E + 05 and, 24 h later, transfected with either GFP or HMOX1$^{GFP}$. After 24 h, cells were exposed to 500 µM $H_2O_2$ in media and counted as in (**d**). For unpaired $t$-test at indicated points relative to 0 µM $H_2O_2$ control; $*p < 0.05$. In all cases, ±refers to with and without, blue refers to WT cells; green refers to $\Delta CHD6$ cells. **f** A model for CHD6 function. (1) ROS such as hydroxyl radicals react with DNA to produce oxidative damage. (2) Oxidative stress suppresses proteasomal degradation of CHD6, stabilizing protein levels, while oxidative DNA breaks elicit PARP enzyme activation, generating PAR polymers in the vicinity of DNA lesions that enable CHD6 relocalization to oxidative DNA lesions. (3) CHD6 contributes to nucleosome compaction and transcriptional responses, which potentially modulate the magnitude of ATM-dependent signaling. (4) CHD6 promotes oxidative stress transcriptional responses (potentially via Nrf2) and, as a consequence, antioxidant proteins such as HMOX1 and TXNRD1 suppress oxidative stress and DNA damage

during oxidative stress. HMOX1$^{GFP}$ was well tolerated, and although growth slowed after GFP or HMOX1$^{GFP}$ transfection, minimal cell death was observed. As with naive cells, A549$^{\Delta CHD6}$ expressing GFP experienced cell growth failure and death after chronic H$_2$O$_2$ exposure; by contrast, A549$^{\Delta CHD6}$ expressing HMOX1$^{GFP}$ were able to withstand oxidative stress to a similar extent as wild-type A549 transfected with either GFP or HMOX1$^{GFP}$ (Fig. 7e). Collectively, these data indicate a substantial deficit in the ability of CHD6 knockout cells to withstand oxidative stress, with CHD6-dependent oxidative stress responses being important for growth and survival following acute exposure and crucial in chronic stress scenarios.

## Discussion

Endogenous and exogenous ROS generate thousands of oxidative lesions per cell on a daily basis. As oxidative genomic damage is mutagenic and potentially lethal, cells must repair injured DNA and/or evade damage induction to survive, function normally and proliferate. Our results indicate that CHD6 may be a primary oxidative DNA damage response factor (Fig. 7f), although at this point it is unclear whether it exerts its effects though (i) direct activity at DNA damage sites, (ii) chromatin compaction regulation, (iii) the antioxidant response transcriptional regulation or (iv) some combination of (i–iii). Our results with HMOX1 expression indicate that CHD6's role in antioxidant regulation is likely a major contributor to its impact on cell growth and survival. As such, CHD6 joins a growing list of chromatin remodelers that respond to DNA damage to promote recovery[46]. During oxidative stress, loss of CHD6 leads to elevated DNA break and PAR induction, increased ROS, reduced antioxidant production, abnormal chromatin status, greater ATM signaling, hypersensitive checkpoint activation and failure to grow and thrive. CHD6 is stabilized during oxidative stress, and relocates to oxidative DNA damage sites. CHD6 is distinguished from other CHD enzymes, such as CHD1, CHD2 and CHD4, in that it appears to have no direct role in the DNA repair pathways examined so far, does not bind non-oxidative DSBs and exacerbates oxidative stress when ablated. CHD6 is also distinct to CHD3 as, although deletion enhances ATM activation, it has little impact on the ATM dependency of DSB repair and is not ultimately lethal, indicative that CHD6 (unlike CHD3)[10] is dispensable for constitutive heterochromatin building or inheritance through DNA replication. Nevertheless, like CHD3 depletion, loss of CHD6 enhances nuclease accessibility to chromatin, raising the possibility that it is either a key part of complexes that regulate linker histone DNA access or is a net chromatin compacter. This latter idea is consistent with evidence that CHD6 binds to human papillomavirus E8-E2C proteins and to repress the E6/E7 promoter[47].

As an alternative explanation to the chromatin hypothesis, it is quite possible that the enhanced ATM signaling seen in ΔCHD6 cells is due to direct ATM activation via the formation of disulfide cross-linkage within ATM dimers driven by elevated ROS levels, even when DSBs are limiting or absent[48], or via SSB-mediated activation as described previously[49]. A central question emerging from our work is how CHD6 can be involved in promoting antioxidant transcription as well as broadly suppressing chromatin accessibility? It is important to note that CHD6, like many chromatin remodeling enzymes, likely contributes to the compaction of some genomic loci, while relaxing others (reviewed in ref. [1]). To clarify this, an important next step will be mapping CHD6 chromatin occupancy across the human genome with and without oxidative stress, as well as identifying interacting partners conferring differential functional specificity. Important first steps in this were taken recently, in a study demonstrating that CHD6

tethers cis-acting CFTR (cystic fibrosis transmembrane conductance regulator) regulatory elements in proximity to a variety of transcriptional complexes important for cell differentiation[50].

Our results suggest that CHD6 also contributes to ROS detoxification, a feasible explanation as to why, mechanistically, CHD6-ablated cells experience elevated oxidative DNA damage (and increased PAR) for a given dose of H$_2$O$_2$. We speculate that CHD6 loss leads to a failure to thrive in an oxidatively stressed environment due to the multiplying effects of reduced antioxidant capacity, greater DNA damage and PAR induction coupled with amplified signaling manifesting together as elevated ATM and checkpoint arrest and/or cell death. This may well explain the phenotype of cerebellar degeneration defects found in CHD6 Δexon12 mice and, in the future, it will be informative to challenge such mice with ectopic oxidative stress to test whether degenerative phenotypes, DNA damage induction, NAD+ depletion (which is tied with steady-state PAR induction) and the related parthanatos cell death process are exacerbated[51].

As we found for CHD6, CHD2 and CHD4 retention at DNA damage is PAR dependent. CHD2 recruitment is conferred by a C-terminal PAR-binding region[3], while CHD4 reportedly interacts with PAR via an N-terminal HMG box-like domain[7,52]. Neither of the CHD2/4 PAR-binding regions are present in the CHD6 polypeptide; rather, we map the PAR-dependent binding region to a small area between the NLS and double chromodomain containing a conserved KR-rich patch. This area is required for the very rapid, early relocalization of CHD6 to sites of DNA damage, with the slower accumulation of the protein then requiring the double chromodomain and other central core domains. Answering whether CHD6 directly binds PAR, as well as elucidating the exact functional role of each central domain, will require this large protein (and mutants thereof) to be purified and biochemically interrogated. It will also be important to further elucidate what CHD6 does at sites of DNA damage as, although mutants defective in DNA damage retention (e.g., ΔCD1+2 mutant) cannot complement the ΔCHD6 phenotype, nucleosome (or other) substrates of CHD6 at oxidative DNA damage need to be identified.

Cancers experience increased proliferation and inflamed microenvironments leading to chronic oxidative stress, and must adapt to survive[53]. Our results fit with the idea that CHD6 promotes growth under adverse oxidative conditions, and might explain why cancers arising from often oxidatively stressed tissues (e.g., colorectal cancer) display CHD6 gene amplification (Supplementary Fig. 4f). Unlike other CHD enzymes known to respond to DNA damage induction, we find that CHD6 protein levels are regulated by a proteasome-mediated degradation process that responds to oxidative stress. This is reminiscent of p53 or Nrf2 upregulation under similar circumstances and, in those cases, ubiquitin-ligases (MDM2 and KEAP1, respectively) are switched off following stimuli to suppress ubiquitin-directed proteasome degradation;[53] however, we have not been able to observe direct ubiquitylation of CHD6 using standard approaches. An alternative possibility is that oxidation of potentially important cysteine or methionine residues in CHD6 might underlie alterations to stability. Ascertaining whether constitutively increased CHD6 expression (in tumors) influences cancer patient survival will be important, as this may consolidate CHD6 as a novel prognostic biomarker or anti-cancer target.

In conclusion, we suggest that CHD6 represents a central player in the oxidative DNA damage response, promoting antioxidant expression and chromatin compaction that, collectively, modulate DNA damage induction and persistence, DNA damage signal transduction and cell cycle checkpoint sensitivity to facilitate cell growth and survival under conditions of acute or chronic oxidative stress.

## Methods

**Reagents and tissue culture.** Phenylmethylsulfonyl fluoride (PMSF), Biotin, Wortmannin (WM), Microcystin-LR (MC-LR), N-ethylmaleimide (NEM), dithiothreitol (DTT), 3-aminobenzamide (3-AB), $H_2O_2$ and streptavidin-agarose were all from Sigma. From Selleck Chemicals we obtained: PARPi AZD2281/Olaparib, used at 2.5 μM; DNA-PKc inhibitor (DPKi) NU7441 (KU57788), used at 10 μM; and bortezomib (BTZ), used at 100 nM. From Tocris we obtained: PARGi PDD00017273, used as indicated, and ATM inhibitor (ATMi) KU-55933, used at 10 μM. IR=gamma rays ([137]Cs) was delivered by GammaCell 1000 Elite (MDS Nordion). A549 (ATCC CCL-185) and HEK293 (ATCC CRL-1573) were cultured in Dulbecco's modified Eagle's medium (DMEM) with 10% (v/v) fetal calf serum (FCS), GlutaMAX (Gibco), penicillin and streptomycin. U2OS-2-6-3 cells obtained from Dr. Susan Janicki (Wistar Institute, USA) are described in ref. [54] and were also cultured in DMEM (as above). U2OS 2-6-3 cells stably expressing ER-mCherry-LacR-FokI-DD obtained from Dr. Roger Greenberg are described in ref. [26] and were induced for 5 h by 1 μM Shield-1 (Clontech) and 1 μM 4-OHT (Sigma). U2OS XRCC1[−/−] cells were obtained from Dr. Keith Caldecott (University of Sussex, UK) and were also cultured in DMEM (as above). All cell lines are tested regularly for mycoplasma contamination and confirmed to be negative. A549 cell identity was confirmed by gene sequencing and karyotyping.

**Immunoblotting (IB) and immunoprecipitation (IP).** For IB or IP, cells were lysed in 5× packed cell volume of ice-cold NETN buffer (150 mM NaCl, 0.2 mM EDTA, 50 mM Tris-HCl pH 7.5 and 1% (v/v) NP-40 detergent) supplemented with 1× protease inhibitor cocktail (Roche), 1 mM DTT, PMSF, 1 μM WM, 10 mM NEM, 5 μM PARGi, 3-AB, and 0.1 μM MC-LR and sonicated (10%, 5 s) and clarified at 10,000 rpm for 10 min. For GFP IP examining interactions with PAR, 2 mg of lysate was incubated with 5 μL packed, equilibrated GFP Trap beads (Chromotek) for 2 h at 4 °C with rotation. For streptavidin-agarose pulldowns, beads were incubated in 50 μL 1× phosphate-buffered saline (PBS) containing either 10 μM free biotin, 10 μM PAR-biotin (Trevigen) or 10 μM H3K9me3-biotin (Epicypher) for 0.5 h at 4 °C, with rotation, before being quenched by incubation with 500 μL of 1 mM TRIS pH 8.8 containing 80 μM free biotin for 15 min at 4 °C, with rotation, and washed 2× with 1 mL NETN buffer. Then, 2 mg lysate was incubated with 5 μL packed beads for 2 h at 4 °C, with rotation. After incubation, all IP pulldowns were washed up to 5× with 0.3 mL NETN. For IB, washed IPs or 25-50 μg lysates (inputs) were incubated at 95 °C for 5 min in 1× Laemmli SDS Sample buffer and resolved by sodium dodecyl sulfate-polyacrylamide gel electrophoresis (SDS-PAGE). Secondary antibodies for IB were anti-mouse/rabbit/goat-HRP from Sigma. Immunolots were visualized by chemiluminescence and either film (Figs. 1a, e–g, 4a) or digital imaging (all other images) using a ChemiDoc (BioRad). Full size, uncropped scans or digital images of immunoblots are available in Supplementary Figs. 5-7.

**Antibodies and immunofluorescence (IF).** The primary antibodies from this study are indicated in Supplementary Table 2, along with the dilutions used. All secondary antibodies for IF were anti-mouse or anti-rabbit IgG coupled to Alexafluor 488 or 594 (Invitrogen Molecular probes; used at 1:800). For widefield microscopy IF, PBS-washed cells were fixed in 3% (w/v) paraformaldehyde (PFA) + 2% (w/v) sucrose for 10 min, permeabilized for 3 min in 0.2% (v/v) Triton X-100 (in PBS) and immunostained for 1 h with primary antibody (diluted in 2% (w/v) BSA in PBS) for 30 min with 1:200 dilutions of secondary antibodies (also in 2% BSA as before). Where indicated, cells were counterstained with 0.1 μg/mL 4′,6-diamidino-2-phenylindole (DAPI) to visualized nuclei and were mounted using Polymount G. Samples were imaged with a Zeiss Axio Observer Z1 platform microscope, with a Plan Apochromat 20×/0.8, an EC Plan Neofluar 40×/0.75 or a Plan-Apochromatin 63×/1.4 (oil immersion) objective and an AxioCam MRm Rev.3 camera. Acquisition and analysis software used was Zen Pro (Zeiss). For confocal imaging, cells were fixed in 3% (w/v) PFA + 2% (w/v) sucrose for 10 min, permeabilized for 5 min in 0.5% (v/v) Triton X-100 (in PBS), and immunostained for 1 h at room temperature (RT) with primary antibody (diluted in 2% (w/v) BSA in PBS). For staining endogenous CHD6, primary antibody staining was carried out at 37 °C for 30 min with 1:400 dilutions of secondary antibodies (also in 2% (w/v) BSA as before). Cells were counterstained with 0.1 μg/mL DAPI to visualize nuclei and were mounted using Fluoromount G (Electron Microscopy Sciences). Samples were imaged with LSM880 Carl Zeiss confocal microscope, with a Plan Apochromat 20×/0.8 NA, an EC Plan Neofluar 40×/0.75 NA or a Plan Apochromat 63×/1.4 NA (oil immersion) objective and a camera (AxioCam MRm Rev.3; Carl Zeiss) or GaAsP or Airyscan detectors (Carl Zeiss). Acquisition and analysis software used was Zen Blue and Zen Black (Carl Zeiss).

**G2/M cell cycle checkpoint IF analysis.** G2/M cell cycle checkpoint IF analysis was carried out exactly as in ref. [41]. Briefly, logarithmically dividing parental and A549[ΔCHD6] cells grown on glass coverslips coated with 0.1% (w/v) gelatin were treated (as indicated) and, at indicated time points, were fixed and immunostained (as outlined above) for H3S10p, 53BP1 and DAPI. Cells were imaged under 10× magnification for H3S10p and DAPI signal, and both total number and strongly H3S10p-positive cells with nuclear morphology indicative of mitosis were scored

for 1200–5500 cells per experimental repeat, with at least 4–6% of overall cell population being mitotic in the untreated condition.

**The 355 nm laser micro-irradiation and live cell imaging.** A549 cells were transfected with 2 μg CHD6 expression construct (as indicated) 16 h before being incubated with 10 μM BrdU (from Sigma) for 16–24 h. Small molecule inhibitors were all added to cells 1 h prior to DNA damage induction. DNA damage tracks were induced in live cells (kept at 37 °C in a humidified environment at 5% $CO_2$) using a 355 nm 5 mW self-aligning solid state diode laser (15 μm/s, 30% power) projected through a EC Plan-Neofluor 40× objective, via a Zeiss PALM MicroBeam laser microdissection module on a Zeiss Axio Observer Z1 platform. Images were captured on an AxioCam MRm Rev.3 camera. Laser irradiation was controlled by RoboSoftware 4.5. Acquisition and analysis software used was Zen Pro (Zeiss). CZI image files were captured every 60 s for the indicated times, and subsequently analyzed using Zen image processing software. For analysis of signal gain, a region of interest was created to cover the area of the laser track and the signal intensity (arbitrary units) within this area was measured over the time course. To measure cell background, a region of interest was cloned and placed next to the region containing the track, providing the signal intensity for background of an area of identical size within the nuclei. The signal intensity of background is then subtracted from the signal intensity of the track, divided by the sum of both regions of interest and presented as a percent gain in signal intensity with the pre-damage (t = 0) representing baseline.

**Site-specific Fok1 and KillerRed DNA damage system.** U2OS 2-6-3 cells stably expressing ER-Fok1-mCherry-LacR-DD (estrogen receptor (ER), destabilization domain (DD))[26] (obtained from Dr. Roger Greenberg, University of Pennsylvania, USA) were induced with 300 nM 4-OHT and 1 μM Shield-I for 5 h. Subsequently, cells were pre-extracted with 0.25% (v/v) Triton X-100 in cytoskeletal (CSK) buffer (10 mM HEPES pH 7.4, 300 mM Sucrose, 100 mM NaCl, and 3 mM $MgCl_2$) for 10 min, fixed with formaldehyde and immunostained with the indicated antibodies. KillerRed cDNA (obtained from Dr. Joachim Goedhart, University of Amsterdam, The Netherlands) was amplified by PCR using the primers: 5′-TCAGCTAGCGTGTACGGTGGGAGGTCTA-3 and 5-AGTTGTACA-CATCCTCGTCGCTACCGATG-3, and inserted into mCherry-LacR-stop using NheI and BsrGI to generate KillerRed-LacR-stop. U2OS 2-6-3 cells were co-transfected with 1.5 μg KillerRed-LacR-stop and 1 μg GFP-CHD6 or GFP-XRCC1. Immediately after transfection, dishes containing cells were wrapped in aluminum foil to prevent KillerRed activation. The induction of ROS by activation of KillerRed was done essentially as described previously[55]. Briefly, 24 h after transfection, cells were exposed to a 24 W Osram dulux white fluorescent bulb for 375 s in a customized stage 15 cm away from the light bulb. The 375 s exposure to light at a rate of 24 W ( = 24 J/s) results in dose of 9000 J. Alternatively, cells were exposed for 188 s (4500 J) or 94 s (2250 J) when indicated. After exposure, cells were immediately fixed for 20 min with 4% formaldehyde. In parallel to KR activation, cells were treated with 5 μM PARGi for 30 min and subsequently exposed to varying concentrations of $H_2O_2$ (0.1, 0.5 or 1.0 mM) for 1 h in DMEM + 10% FCS to compare $H_2O_2$-induced and KR-induced PAR, γH2AX or XRCC4 recruitment.

**Plasmids, siRNA and transfection.** The GFP-tagged CHD4 construct is described in ref. [3]. GFP-C3-PARP1 was a gift from Valerie Schreiber as described in ref. [56]. GFP-tagged XRCC1 was obtained from Keith Caldecott (University of Sussex, UK). The GFP-tagged HMOX1 plasmid (pCX-HO1-2A-EGFP) was a gift from Roberto Giovannoni (Addgene plasmid # 74672)[57]. CHD6 expression constructs (CHD6[GFP], CHD6[FLAG], CHD6[mPLUM]) were made by cloning in full-length wild-type human CHD6 cDNA (accession NM_032221; NP_115597) into either a pEGFP-C2 backbone, pCMV6-Entry-FLAG or mPlum-C1 (a gift from Michael Davidson (Addgene plasmid # 54839)[58] backbone, all under cytomegalovirus (CMV) promoters (Clonetech). Where indicated, GFP alone refers to empty pEGFP-C2 vector. Plasmid transfection was achieved using PolyPlus (VWR) according to the manufacturer's protocol. siRNA-mediated knockdown was also achieved with PolyPlus transfection using 100 pmol of siRNA duplexes per $2 \times 10^5$ cells. The siRNA to CHD2 and Ku80 (and scrambled controls) in Fig. 5g were as described in ref. [3]. Other siRNAs were Stealth™ siRNA oligos from Invitrogen used as a 1:1 pool of A+B sequences as indicated in Supplementary Table 3. GFP-tagged CHD6 plasmid mutations (including amino acid changes, nucleotide positions and changes) are indicated in Supplementary Table 4.

**Quantitative PCR.** Cell pellets were lysed with Trizol reagent (Ambion) to extract RNA. cDNA was synthesized with 1 μg/μL of RNA using Superscript II reverse transcriptase (Invitrogen). qPCR analysis was performed using SYBR green (Life Technologies) to determine Nrf2 target mRNA expression. Conditions were loaded in triplicate and for analysis, the CT values from each repeat were averaged. The averaged CT values for each condition were normalized to the respective GAPDH for loading control to obtain the ΔCT value. To obtain the ΔΔCT, the values for each treated condition was normalized to the parental NT. The ΔΔCT values were then expressed as fold change in expression relative to NT averaged over three independent repeats. Primers sequences are indicated in Supplementary Table 5.

**CRISPR-based gene editing**. CRISPR guide sequences targeting CHD6 exon 3 were designed using the online CRISPR design tool at (crispr.mit.edu): CHD6 sgRNA: AA ACGTATACTGCTGAAGAGGAAGC. Guide sequences were integrated into the pXPR_001 plasmid containing Cas9[59]. Constructs were transfected into parental A549 cells and subjected to puromycin (1 mg/mL) selection for 7 days. After this, polyclonal cells were assayed for CHD6 expression and subjected to single cell cloning by serial dilution. Single cell clones were screened by immunoblotting and clones positive for loss of CHD6 expression were identified, expanded and confirmed by PCR analysis using primers adjacent to the CHD6 sgRNA sequence: CHD6 exon3, forward: 5'-GGAATTCCACTCCCCAATGTCTGATGC-3'; CHD6 exon3, reverse: 5'-CGGGATCCTTGTGCTCCTTGGCCTTCTT-3'. DNA sequencing confirmed that both alleles of CHD6 in the parental A549 line conformed to the GenBank wild-type CHD6 nucleotide (nt) sequence, while multiple (eight) independent runs indicated that 100 nt were deleted in both alleles of the ΔCHD6 A549 line, specifically nt 186–285 (corresponding to amino acids 63-95) of the third exon, with the remainder of the gene sequence conforming to wild type.

**Micrococcal nuclease (MNase) chromatin relaxation assay**. Performed exactly as in Klement et al.[10]. Briefly, nuclei were isolated from confluent cultures of parental and ΔCHD6 A549 cells. For MNase (Nuclease S7, from Roche) digestion, 75 μL nuclei was used per time point, with the addition of 1 U/μL MNase and incubation at 25 °C for indicated times. Then, 75 μL reactions were quenched with the addition of 1.5 μL 0.5 M EDTA. Protein was digested with 1 mg/mL Proteinase K (Sigma) in 5% (w/v) SDS for 30 min at 37 °C, extracted with phenol/chloroform, washed with diethyl ether and DNA precipitated with the addition of ethanol to 75% and incubation overnight at −20 °C. Then, 2.5 μg of rehydrated DNA was resolved on a 1.2% (w/v) agarose gel run in 1× TAE buffer.

**Clonogenic survival analysis**. For acute $H_2O_2$ exposure, 300 cells were seeded in duplicate into 6 cm dishes and allowed to attach for 8 h. Media were then removed, cells were washed with 1× PBS and exposed to $H_2O_2$ in ice-cold 1× PBS at RT before being rinsed in 1× PBS and placed back into conditioned media. For chronic $H_2O_2$ exposure, confluent cells were exposed to $H_2O_2$ in media for 48 h, with fresh $H_2O_2$ added daily before being split and 300 cells seeded into 6 cm dishes containing media with more $H_2O_2$. In all cases, cells were fixed 10 days later in 3% (v/v) acetic acid, 8% (v/v) methanol and stained with crystal violet (0.2% (w/v) crystal violet, 4% (w/v) PFA in PBS). Colonies were counted expressed as a percentage of untreated.

**Alkaline comet assay**. The day before, 0.8 % (w/v) in PBS agarose (Invitrogen) plugs were prepared on frosted slides (VWR). The lysis buffer (5 M NaCl, 0.5 M EDTA, 1 M Tris pH 10; pH 10 with 5 M NaOH) and electrophoresis buffer (10 M NaOH, 0.5 M EDTA) were also prepared the day before. Parental and ΔCHD6 A549 cells were collected, washed once with 1× PBS, counted and diluted to $2 \times 10^5$ per mL. Then, cells were resuspended in PBS containing the indicated dose of $H_2O_2$, returned to the incubator and 1 mL of cells was removed per time point. After treatment, cells were washed 2× with 1× PBS and the pellet collected and resuspended in 1 mL of PBS. For IR treatment, the cells are resuspended in cell media, irradiated with indicated dose, returned to the incubator and 1 mL of cells removed per time point. Cells were washed and prepared as above. Using 1.2% (w/v) low-melting agarose (Invitrogen) in 1× PBS, 200 μL of cells were placed in a fresh tube to which 200 μL of warm low-melting agarose was added, mixed and 200 μL was quickly pipetted on top of pre-set agarose plugs under a cover-slip. The plugs containing cells were left in the dark at 4 °C to set for 30 min. For lysis, coverslips were removed and slides placed in a black container in a dark cold room and incubated in lysis buffer (supplemented with 1% (v/v) DMSO and 1% (v/v) Triton X-100) for 1 h. Cells were washed 3× in cold $dH_2O$ then slides placed in an electrophoresis chamber and covered in electrophoresis buffer (supplemented with 1% (v/v) DMSO) for 45 min in the dark to neutralize pH. Electrophoresis was carried out at 25 V for 25 min, comets neutralized by submerging plugs in 0.4 M Tris pH 7.0 until ready to score. Immediately prior to scoring, slides were stained with 1:10,000 dilution of SYBR green supplemented with 4 μg/μL of antifade (stock at 11 μg/μL) for 5–10 min at RT in dark, removed and scored immediately. Comet scoring was performed on a Zeiss Axiovision microscope with a camera associated with Comet Assay IV software (Perspective Instruments). A total of 100 comets were scored per condition, per repeat the data expressed as the average comet tail moment on a scatter plot.

**Cell growth analysis**. Exactly 1E+05 cells were seeded in 3 cm dishes, and allowed to recover for 24 h. For acute $H_2O_2$ exposure, media were then removed, cells were washed with 1× PBS and exposed to $H_2O_2$ in ice-cold 1× PBS at RT before being rinsed in 1× PBS and placed back into conditioned media. For chronic $H_2O_2$ exposure, cells were exposed to $H_2O_2$ in media, with fresh $H_2O_2$ added daily for the duration of the experiment. Each day, cells were rinsed in 1× PBS, resuspended using trypsin-EDTA and viable cells were enumerated using a Moxi-Z cell counter.

**Oxidative stress assay**. $H_2DCFDA$ (Thermo Fisher, D399) was prepared as a stock of 20 mM in DMSO. Conditioned media were removed from cells and stored at 37 °C; cells were washed in 1× PBS and placed into phenol red and serum free media (DMEM, from Gibco) containing 20 μM $H_2DCFDA$ for 1 h at 37 °C. Cells were then washed in 1× PBS and returned to conditioned media ± 500 μM $H_2O_2$ for 20 min. Cells were then harvested by scraping into ice-cold PBS, washed once with 1× PBS+0.0025% (w/v) Trypan blue to quench extracellular $H_2DCFDA$. 1E +06 cells were then subject to FACS analysis (University of Calgary FACS facility). For addback experiments, A549^ΔCHD6 cells were transfected with 4 μg of the CHD6-mPlum expressing construct 24 h prior to $H_2O_2$ exposure. For analysis, percent of P3-positive FITC cells was used as an indication of DCF intensity. For delta cells complemented with CHD6-mPlum, FITC-positive cells were quantified as a subset of mPlum expressing cells. The gating strategy applied for this experiment is indicated in Supplementary Fig. 8a.

**Multiplexed DNA repair assays**. FM-HCR assays were carried out in A549 and A549^ΔCHD6 cells, as previously reported[37,38]. A549 and A549^ΔCHD6 were maintained in culture under ambient culture conditions (37° C/5% $CO_2$). Cells were seeded 2 days prior to harvesting for transfection (cells were harvested for transfection at ~85% confluency). Ambient culture conditions for FM-HCR plasmid assays were compared to low oxygen culture (37 °C/3% $O_2$). For 3% oxygen culture, A549 and A549^ΔCHD6 cells were equilibrated at 3% $O_2$ for 4 h and collected at the same time of ambient cell harvesting. A Neon Transfection System (Thermo Fisher Scientific) was used to transfect 500,000 cells using a 10 μL tip two 1200 V 30 ms pulses. Cells were transfected with reporter plasmids with DNA lesions or corresponding undamaged reporter plasmids (see Supplementary Table 6)[60]. After transfection, A549 and A549^ΔCHD6 cells were seeded into 12-well plates containing 2 mL of media per well, and were placed into either ambient or 3% $O_2$ culture conditions. At 24 h post transfection, cells were harvested and analyzed for DNA Repair Capacity (DRC) by flow cytometry. Fluorescent protein signal was quantitated for each respective plasmid for 15,000 cellular events per replicate ($n = 3$). Percent reporter expression for fluorescent proteins was quantified as previously described[36,37]. The gating strategy applied for this experiment is indicated in Supplementary Fig. 8b.

**Mass spectrometry analysis of histones**. Parental or delta CHD6 cells, $1 \times 10^7$ cells were collected and lysed in hypotonic lysis buffer prior to acid extraction of histones with 0.4 N $H_2SO4$ based on a previously descried protocol[61]. Histone concentration was determined by running histone solution (dissolved in $dH_2O$) on an SDS-PAGE gel and staining with Coomassie Brilliant Blue. Histone purification and analysis were performed as previously described, with minor modifications[62]. Briefly, histones were isolated using acid extraction and the trichloroacetic acid (TCA)-precipitated histones were resuspended in $ddH_2O$. Purity was confirmed by SDS-PAGE. The pH was adjusted to 8.0 by the addition of $NH_4HCO_3$ to a final concentration of 50 mM. Subsequent derivatization and digestion was performed in two independent replicates for each cell line. Histones were propionylated in two rounds exactly as described previously[63] and subsequently digested with trypsin at an enzyme-to-substrate ratio of 1:20. Digestion was carried out overnight at 37 °C. After digestion, the derivatization reaction was repeated to label peptide N termini. Samples were desalted using C18 ZipTips prior to nano liquid chromatography-tandem mass spectrometry (LC-MS/MS) analysis. All histone samples were analyzed on an Orbitrap Velos mass spectrometer coupled to an EASY-nLC 1000 system (Thermo Fisher Scientific, Bremen, Germany) equipped with a Nanospray Flex ion source as it has been described elsewhere[64]. The flow rate was set to 300 nL/min. All four samples were analyzed in triplicate. A gradient consisting of solvent A (97% $H_2O_{dd}$, 3% ACN, 0.1% FA) and solvent B (97% ACN, 3% $H_2O_{dd}$, 0.1% FA) running linearly from 2 to 28% B within 40 min, followed by an increase to 42% B within 12 min was applied. For C18 column regeneration, the gradient was ramped up to 95% B within 5 min and kept at 95% B for 5 min. Data were acquired using data-dependent MS/MS mode as described in ref. [62] with minor modifications. In brief, for the first 18 min each high-resolution precursor ion scan in the Orbitrap ($m/z$ 300–1250, $R = 60,000$) was followed by high-resolution product ion scans (isolation window 2 Th) after higher-energy collisional dissociation at 35% normalized collision energy (NE). The resolution was set to 7500. The 8 most abundant signals with a charge state ≥2 and a minimal intensity of 15,000 were selected for fragmentation, followed by dynamic exclusion for 30 s. For the next 20 min, MS1 data were acquired with the same settings, but an inclusion list was further used containing five ions, which represent isobaric and co-eluting histone peptides (H3 9-17aa 1 acetyl ($m/z$ 528.30), H3 18-26aa 1 acetyl ($m/z$ 570.84), H4 4-17aa 1 acetyl ($m/z$ 768.95), H4 4-17aa 2 acetyls ($m/z$ 761.94), H4 4-17aa 3 acetyls ($m/z$ 754.93)). Peptides were isolated and dissociated with the same settings as described above. Besides repeatedly triggering these five ions for MS2 over this time frame, a Top 3 method was additionally included to fragment other eluting peptides in this window. For the last 22 min, the same Top 8 acquisition strategy was chosen as for the first 18 min of the LC-MS/MS run. All data were analyzed using EpiProfile 2.0 with standard settings, as described previously[65]. All RAW data are publicly available via Chorus (https://chorusproject.org/pages/index.html; ID 1497).

**Statistical analysis**. Statistical analysis was performed using Prism software. Data represented as mean + s.d. or s.e.m., as indicated. Significant differences were calculated using Student's *t*-test where indicated. For laser micro-irradiation data, differences in the overall recruitment to damage sites was determined by a two-way analysis of variance (ANOVA; multiple conditions with multiple time points) and subsequently analyzed by Tukey's test to determine differences at individual time points. For all tests, *P* values are as follows: NS non-significant (>0.05); *statistically significant (<0.05); **statistically significant (<0.01); ***statistically significant (<0.001); ****statistically significant (<0.0001). Experiments were repeated a sufficient independent number of times to ensure reproducibility of results. See specific figure legends and Methods sections for details.

**Reporting summary**. Further information on experimental design is available in the Nature Research Reporting Summary linked to this article.

### Data availability

The data sets generated and/or analyzed during the current study are available from the corresponding author on reasonable request. The raw data from the Mass Spectrometry analysis of histones are publicly available via Chorus (https://chorusproject.org/pages/index.html; ID 1497).

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

## Acknowledgements

We thank Brielle Cram, Connor Pettem and Luc Boutin for preliminary work; Dr. Anne Vaahtokari and the Charbonneau Institute Microscopy Facility for imaging assistance; Sophie Briggs and the lab of Dr. Jennifer Cobb for qPCR advice; Justin Simms for CRISPR advice; Yaping Yu for gene sequence analysis; Keith Caldecott for comet assay advice and Roger Greenberg for providing the FOK1-LacR cells. MSL is supported by a VENI fellowship (ALW-863.11.007) and VIDI fellowship (ALW-016.161.320) from the Netherlands Scientific Organization (NWO). HvA is supported by an ERC Consolidator grant from the European Research Council. CGP and the ZDN lab is supported by SBDR P01 CA092584. CUS and the DCS lab are supported by an NSERC RGPIN-2017-04879 grant. SM was supported by an Achievers in Medical Sciences award. AAG is the Canada Research Chair for Radiation Exposure Disease and this work was undertaken, in part, thanks to funding from the Canada Research Chairs program. The AAG laboratory is supported by the Canadian Institutes of Health Research.

## Author contributions

In the A.A.G. lab, A.A.G. and S.M. designed the study and contributed to all figures, N.D.B. (co-supervised by A.A.G. and J.A.C.) provided data in Figs. 1, 4, 5 and F.K.T.S. provided data in Fig. 1 and Supplementary Fig. 1. M.S.L. and H.v.A. provided data in Fig. 2. C.G.P. with Z.D.N. and C.U.S. with D.C.S. provided data in Fig. 5. S.F. generated CHD6 constructs. All authors contributed to manuscript preparation.

## Additional information

**Competing interests:** The authors declare no competing interests.

