## [Peer Review File · Nature Communications]

Reviewers' comments:

Reviewer #1 (Remarks to the Author):

This paper provides novel insight into the dual role of CHD6 in regulating the cells response to oxidative stress. A key novel insight is that CHD6 can be both recruited to sites of damage as well as play a key role in transcriptional regulation of the anti-oxidant response. A component of this regulation is the rapid stabilization of CHD6 in cells undergoing oxidative stress. This paper provides some of the first data linking CHD6 to oxidative stress, and may have implications for how deregulation of CHD6 contributes to tumor progression. The data is strong and the overall analysis, including statistical analysis, is excellent. The demonstration that CHD6 is stabilized after ROS and the comparison of FOK1 vs LacR-KR to show that it is oxidative damage rather than e.g. DSBs is both clever and strong. Further, there is a clear link between loss of CHD6 and an increase in PAR levels following ROS damage. The data showing that CHD6 may regulate expression of anti-oxidant genes is strong and provides a mechanistic basis to explain CHD6 function. The experiments to analyze the CDs of CHD6 are interesting, but will require significant work to fully validate. Further, the data showing increased MNase sensitivity after CHD6 loss are potentially important for understanding how increased damage and ROS sensitivity may occur in the CHD6 knockout cells, although how this fits with the transcriptional function of CHD6 is not clear. Overall, this paper provides strong support for a novel, dual role for CHD6 in both regulating repair at sites of damage and in upregulating genes responsible for promoting ROS metabolism and therefore limiting DNA damage during acute or chronic ROS exposure.

Major points:

- For the LacR-KR system, it would be anticipated that there would be a mix of single and DSBs, base damage etc. What level of damage was created by the given light exposure? That is, could there be so much damage that the DNA is largely fragmented in this region and this prevents effective recruitment of repair proteins? Can it be shown that e.g. Ku70/80 or other key DSB proteins are also excluded from the KR region? And varying the light "dose" to create different levels of damage?
- Fig S1G indicates that CHD6 overexpression suppresses PARylation. How does this alter the interpretation of the experiments showing that overexpressed CHD6 is recruited to laser stripes in a PARP1 dependent manner? Does CHD6 overexpression limit either stripe intensity or PARP1 recruitment to laser stripes? Or alter PARP1-3 expression levels?
- The affinity experiments in figure 2G need additional experiments. Biotin peptides with non-methylated H3 and H3 methylated at other sites (e.g. K27) are needed. In addition, if the CDs on CHD6 bind both PAR and H3K9me3, competition experiments with the H3K9me3 peptide should be used to determine if it releases CHD6 from the PAR-biotin. Using cell extracts is also problematic as it cannot be determined if CHD6 is binding directly or indirectly (via another protein) to the affinity resin. In general, these experiments are quite limited and can't be used to determine e.g. differential binding to PAR vs H3K9me3, since the binding affinity is not known. Purified proteins would allow for exact determination of binding affinity and stoichiometry of interaction.
- In figure 3D, does CHD6 loss in the absence of PARGi also cause an increase in cellular PAR?
- The conclusion on lines 165-168 is a bit strong, since the CD still gets to laser stripes (figure 2F), so the CD is only a contributor to recruitment, not the sole mechanism.
- More detailed experiments to distinguish between the 2 proposed functions of CHD6 (direct impact on repair + transcriptional regulation) are needed. For example, do PARP inhibitors (which block CHD6 recruitment to laser stripes) also block transcriptional upregulation of genes in figure 3? Does overexpression of e.g. HMOX1 or other genes rescue loss of CHD6? Can CHD6 be recruited to these genes in response to oxidative damage (i.e. ChIP)?

Minor points:

- Figure 1 might be easier to read if 1B and 1D were moved to supplementary data.
- Quantitation for the data in figs 1E and 1F is needed (% gH2Ax stripes with CHD6 or 53BP1 stripes)
- The statement from figure 1F that 35BP1 and CHD6 only partly overlap is not readily seen in the

image supplied. Quantification of this would help (more images etc).

- A reference for the statement (line 116) that "XRCC1 is a major PAR-binding platform for recruiting DNA damage...") is needed.
- Does deletion of CHD6 lead to upregulation of other CHD family members?
- In figure 3, the differences in PAR are only seen at 1mM H₂O₂, which is very high! Can this be repeated at lower H₂O₂ concentrations?

Reviewer #2 (Remarks to the Author):

In this study Moore and colleagues explore the possible role of CHD6 on the cellular response to oxidative damage. They observe that CHD6 levels are stabilized upon increasing oxidative environment and that CHD6 is actively recruited to sites of oxidative damage in a PAR-dependent manner. In fact this recruitment is, at least, partially dependent on the double chromodomain of CHD6 (interestingly CHD6 double chromodomain binds PAR) and independent of the SSB repair scaffolding protein XRCC1. Although Δ CHD6 cells accumulate higher levels of DSBs in the presence of H₂O₂, when studying the repair efficiency of different DNA lesions no significant repair defect is detected. Curiously, CHD6 ablation promotes higher levels of ROS and oxidative damage-induced PAR. Authors find an explanation with the correlation between CHD6 absence and the downregulation of cellular antioxidant regulators. On the other hand, CHD6 regulates globally chromatin compaction resulting in ATM hyperactivation in the presence of low number of DSBs and increased G2/M checkpoint arrest in the presence of oxidative damage. Overall, CHD6 ablation confers sensitivity to oxidative damage.

This is a very wide characterisation of the role of CHD6 on the cellular response to oxidative DNA damage. There are many interesting observations and most of conclusions are properly supported by experimental data. However, some key questions remain unanswered and the final model seems too speculative.

Major comments

1. It has been shown by many labs that PAR-dependent recruitment of SSB repair factors (including XRCC1) occurs within few seconds. That is much quicker than the recruitment of CHD6 shown in Figure 2 and is in agreement with an XRCC1-independent recruitment of CHD6. However, since parylation occurs so quick, and since CHD6 binds PAR, why is CHD6 not recruited as quick as other PAR-binding factors? And why does it remain for so long? This result would support an indirect role of PAR in CHD6 recruitment or even a retention rather a recruitment one. Authors should clarify or discuss further this fact. FRAP experiments might help.
2. Although laser tracking looks convincing all experiments in Figure 2 should be quantified. A representative image of a single cell is not informative enough when comparing knockdowns (Figure 2D). In the case of Δ Chromo recruitment (Figure 2F), Control plot seems to be the same one than in Figure 2B. If that's the case authors should clarify if both experiments were performed at the same time. Since no biological replicates are shown and laser tracks usually exhibits broad experimental variation controls should have been performed in parallel. If those tracks were not done at the same time, the experiment should be repeated and a proper control included.
3. Due to the high ratio SSB:DSB induced by ROS is quite surprising that authors do not include SSB measurement but only DSBs markers such as 53BP1 and γ H2AX foci. DSB observed in laser tracks are probably the consequence of high power used and therefore coincident SSBs. In addition, Figure 1H suggests DSB-independent recruitment of CHD6. A rational quantification of SSB formation and repair in the absence of CHD6 should be included in this work. Alkaline comet assays are likely the best technical approach for that purpose. Could differences in SSBs explain the earlier activation of ATM and G2/M checkpoint rather the chromatin compaction status?

4. Since FM-HCR is a plasmid-based DNA repair assay, it does not seem to be the best method for measuring chromatin remodellers dependent-repair. Authors could have used CHD1-4 knockdowns as positive controls. Is true that complementary data for studying DSB repair is shown but there is not supportive data in the case of MMR, NER and BER. Again, SSB induction and repair should be checked.

5. To confirm the role of CHD6 in the increase of oxidative stress through HMOX1 and TXNRD1 downregulation authors should actually show that CHD6 is relocalised to these genes upon oxidative stress. They should also try to complement DCF assay (Figure 3E) with CHD6 catalytic mutant and overexpressing NRF2.

Specific points

1. In general, when performing statistic analysis non-significant differences should be indicated as well as significant ones are. Lack of these indications makes reader understand that some important pairs have not been tested.

2. Authors should quantify changes in the levels of other CHD proteins, especially in CHD2 blot, that seems overexposed, making not obvious that no significant changes occur. Figure 1B.

3. Authors should indicate either in the figure legend or in experimental procedures what is the control gene/RNA they use for relative quantification in all RT-PCR experiments.

4. Authors associate CHD6 levels with oxidative stress. Interestingly they show later that CHD6 is only recruited to ROS-induced DNA lesions. How are CHD6 levels regulated upon a non-oxidative source of DNA damage?

5. Figure 2F. Imperfect overlap between CHD6 and 53BP1 seems anecdotal and not very convincing.

6. Add stats to figure 3E.

7. The statistic test used in Figure 2B should be indicated in figure legend, and be applied to Figure 2F.

8. Line 125. "via a potentially methylated PAR-binding partner" seems speculative.

9. Proliferation experiments are not very informative and could be moved to supplementary. In special Figure 6B, in which Control cell number saturates after 4 days. Clonogenic survival experiments are robust enough to support their conclusions. Authors should include statistic analysis (two-way ANOVA) on cell survival.

10. Authors should discuss possible link between progressive ataxia in CHD6 Δ exon12 mice and SSB repair. Since there is an increased PAR induction in the absence of CHD6, do authors detect depleted NAD⁺ levels? This possibility is quite relevant and should be at least discussed. In fact, the second time authors cite reference 19 (line 52) that citation should be substituted by a reference to McKinnon or Caldecott labs where this SSB repair defects-dependent ataxias have been broadly documented.

11. Figure 1E. Endogenous CHD6. Square marking region selected for expanded image is moved.

12. Sections in results are incorrectly numbered.

13. Line 234. is referred to Figure 5A bottom panel rather than 5B.

Rebuttal Comments in **BLUE FOR REVIEWER #1** and in **ORANGE FOR REVIEWER #2**

Reviewer #1 (Remarks to the Author): This paper provides novel insight into the dual role of CHD6 in regulating the cells response to oxidative stress. A key novel insight is that CHD6 can be both recruited to sites of damage as well as play a key role in transcriptional regulation of the anti-oxidant response. A component of this regulation is the rapid stabilization of CHD6 in cells undergoing oxidative stress. This paper provides some of the first data linking CHD6 to oxidative stress, and may have implications for how deregulation of CHD6 contributes to tumor progression. The data is strong and the overall analysis, including statistical analysis, is excellent. The demonstration that CHD6 is stabilized after ROS and the comparison of FOK1 vs LacR-KR to show that it is oxidative damage rather than e.g. DSBs is both clever and strong. Further, there is a clear link between loss of CHD6 and an increase in PAR levels following ROS damage. The data showing that CHD6 may regulate expression of anti-oxidant genes is strong and provides a mechanistic basis to explain CHD6 function. The experiments to analyze the CDs of CHD6 are interesting, but will require significant work to fully validate. Further, the data showing increased MNase sensitivity after CHD6 loss are potentially important for understanding how increased damage and ROS sensitivity may occur in the CHD6 knockout cells, although how this fits with the transcriptional function of CHD6 is not clear. Overall, this paper provides strong support for a novel, dual role for CHD6 in both regulating repair at sites of damage and in upregulating genes responsible for promoting ROS metabolism and therefore limiting DNA damage during acute or chronic ROS exposure. **We thank the reviewer for their support and enthusiasm for our work!**

Major points:

1. For the LacR-KR system, it would be anticipated that there would be a mix of single and DSBs, base damage etc. What level of damage was created by the given light exposure? That is, could there be so much damage that the DNA is largely fragmented in this region and this prevents effective recruitment of repair proteins? Can it be shown that e.g. Ku70/80 or other key DSB proteins are also excluded from the KR region? And varying the light “dose” to create different levels of damage?

We agree that there will be mix of DNA damage types in the LacR-Killer Red system, and now include new experiments (Figs 2C-H) addressing the relative amount of DNA damage relative to hydrogen peroxide. We find that DNA damage marker levels are approximately the same as after 1 hr in 1 mM H₂O₂ – a dose comparable to our other experiments and not a dose that precludes effective recruitment of repair proteins. To demonstrate this, we now show XRCC4 retention at all sites – this core NHEJ protein represents ligation complexes successfully forming at these sites of damage, and its presence argues against a fragmentation effect precluding effective recruitment of repair proteins. This was comparable to XRCC1 and the formation of PAR at the same sites. In further support of this, we measured DAPI staining through the KillerRed (KR) signal at the array (see figure to

right), and also observed no alteration in signal, again suggestive that the DNA in that area is not significantly altered compared to the surrounding area. These new experiments further consolidate the validity of our observations and we thank the reviewer for the helpful suggestions. As the reviewer also suggested, we tried various light exposures ranging from 94-375 seconds, however it seems that this system is essentially “on/off” and a dynamic range of doses to generate different levels of damage is not feasible (See Fig 2).

2. Fig S1G indicates that CHD6 overexpression suppresses PARylation. How does this alter the interpretation of the experiments showing that overexpressed CHD6 is recruited to laser stripes in a PARP1 dependent manner? Does CHD6 overexpression limit either stripe intensity or PARP1 recruitment to laser stripes? Or alter PARP1-3 expression levels?

To address this, we were able to examine PARP1 levels and recruitment to laser induced DNA damage in the CHD6 deleted cells (see Figure S2E-F) and found no significant difference – the expectation being that, if the reviewer’s concern was correct, PARP1 recruitment or expression levels might have been elevated in the absence of CHD6. The data indicates this is not the case. Please also note that the over-expression effect seen in the original Fig S1G was in HEK293 cells, which were used for those more in vitro experiments for the very reason they massively over-express proteins (making them useful as fodder for IPs). All laser experiments were done in A549 cells which express much more modest amounts of CHD6, and only cells with modest expression and normal nuclear morphology are chosen for laser microirradiation. Our data in Figure 4D indicates that while CHD6 can mildly (about 25%) suppress PAR induction in wildtype A549 cells after H2O2 treatment, it does not 100% ablate PAR formation. Our lasers locally produce 8 Gy IR worth of DNA damage, a level we know produces PAR even when CHD6 is overexpressed in these cells; thus, we are confident that this is not a major issue with the interpretation of our laser experiments.

3. The affinity experiments in figure 2G need additional experiments. Biotin peptides with non-methylated H3 and H3 methylated at other sites (e.g. K27) are needed. In addition, if the CDs on CHD6 bind both PAR and H3K9me3, competition experiments with the H3K9me3 peptide should be used to determine if it releases CHD6 from the PAR-biotin. Using cell extracts is also problematic as it cannot be determined if CHD6 is binding directly or indirectly (via another protein) to the affinity resin. In general, these experiments are quite limited and can’t be used to determine e.g. differential binding to PAR vs H3K9me3, since the binding affinity is not known. Purified proteins would allow for exact determination of binding affinity and stoichiometry of interaction.

Our new data functionally dissecting the PAR-dependent recruitment region of CHD6 (data in Fig 3) has altered our original conclusions surrounding these experiments and the role of the double chromodomain. We no longer suggest that the double chromodomain binds to PAR, and indeed we find that its role is to support the later retention of CHD6 at sites of damage and that this process occurs subsequent to the initial PAR-dependent, fast and early recruitment that we now know requires the region mapping to 171-231. We agree with the reviewer that using cell extracts is of limited use, as it cannot provide information on direct binding. Recognizing this, and in the interest

of limited space, we have removed much of that original data and consign the remaining in vitro examination of the wildtype CHD6 interacting with PAR-biotin in extracts to the supplementary data. With regards to the last point, we are working on obtaining purified CHD6 useful for such assays, but that alone would represent a totally distinct body of work far too large to include here. It is our hope that the reviewer will agree that our new data with the seven additional mutants and narrowing the PAR-dependent recruitment region to a conserved 60 aa area will compensate for this, and address the concerns. As a side note, we are currently trying to obtain a point mutant of the conserved amino acids in this area – but this will take more time than is realistic for the duration of the revisions, as CHD6 is a very large cDNA that is not trivial to clone. We aim to include that mutant and analysis in a follow up study at a future date.

4. In figure 3D, does CHD6 loss in the absence of PARGi also cause an increase in cellular PAR? **Yes it does, although in the absence of PARGi all signals are very weak and on the border of significance and detection. We have added this data in Fig S3C.**

5. The conclusion on lines 165-168 is a bit strong, since the CD still gets to laser stripes (figure 2F), so the CD is only a contributor to recruitment, not the sole mechanism.

This conclusion has been removed and indeed entirely re-interpreted based on our new data in Figure 3.

6. More detailed experiments to distinguish between the 2 proposed functions of CHD6 (direct impact on repair + transcriptional regulation) are needed. For example, do PARP inhibitors (which block CHD6 recruitment to laser stripes) also block transcriptional upregulation of genes in figure 3? Does overexpression of e.g. HMOX1 or other genes rescue loss of CHD6? Can CHD6 be recruited to these genes in response to oxidative damage (i.e. ChIP)?

We did the experiment examining whether PARPi impacts the transcriptional upregulation of HMOX1 and TXNRD1 in wildtype cells and found no effect. Thus, we conclude that the PARP-dependent actions of CHD6 (or indeed any pathway) are not directly regulating the antioxidant transcriptional response. This data is in Fig S4A.

We have been trying to perform CHD6 ChIP for some time. There are no available ChIP compatible antibodies to endogenous CHD6. We have been working through conditions using transiently expressed GFP-tagged CHD6 and using GFP-TRAP beads to pull down – this is starting to work, and

we indeed observe a slight (not yet significant) enrichment of HMOX1 and TXNRD1 promoter DNA in our CHD6 ChIP. That preliminary data is shown below for convenience.

However, all attempts at seeing an enrichment after oxidative stress have so

far failed. We have several ideas as to why this might be (including that oxidized DNA, to which we find CHD6 may have a preference for, does not perform well in a PCR), but at this time do not have a satisfactory explanation. This will likely take some time to resolve, is likely a purely technical issue, and hope the reviewer forgives this given the balance of successful data already outlined to address this concern. The good news is that we were able to successfully perform the suggested experiment over-expressing HMOX1 in wildtype and CHD6-deleted cells, examining the PAR induction phenotype. Indeed, in cells with significant HMOX1 expression (demarcated by GFP signal) we see a suppression of the hyper-induced PAR signal seen in the CHD6 mutant cells. This data is in Fig 5A.

Minor points:

7. Figure 1 might be easier to read if 1B and 1D were moved to supplementary data. **We are showing new experiments for both of these figures, some of which is to address the concerns of the second reviewer. We have also moved a lot of data around and generally improved the presentation of data to address this concern regarding readability.**
8. Quantitation for the data in figs 1E and 1F is needed (% gH2Ax stripes with CHD6 or 53BP1 stripes). **Figure 1F has been removed altogether, and the quantification of other data is now shown in new Figure 1I, including additional data comparing CHD6 recruitment kinetics to XRCC1 and PARP1.**
9. The statement from figure 1F that 35BP1 and CHD6 only partly overlap is not readily seen in the image supplied. Quantification of this would help (more images etc). **We agree and removed this data as it was of only minor importance.**
10. A reference for the statement (line 116) that “XRCC1 is a major PAR-binding platform for recruiting DNA damage...”) is needed. **Added.**
11. Does deletion of CHD6 lead to upregulation of other CHD family members? **Great suggestion. It does not, there is no change in other CHDs examined. Please see the new data in Fig S2G.**
12. In figure 3, the differences in PAR are only seen at 1mM H2O2, which is very high! Can this be repeated a lower H2O2 concentrations? **So we actually had a 0.1 mM treatment in the original submission, in what is now Fig 4D. We do still see the differences in PAR (between wildtype and CHD6 deletion mutant) at that lower dose, it is just much more obvious at 1 mM. Please note that this is being added to media, not to a PBS-mediated treatment modality. This is so that cells are metabolically active to produce DNA damage signalling during treatment. In that case, a great deal of the peroxide actually is absorbed by the media itself, functionally lowering the dose that has an effect on the cells. In our hands, 1 mM in media is similar to about 0.2-0.3 mM in a PBS solution in terms of DNA damage induction.**

Reviewer #2 (Remarks to the Author): In this study Moore and colleagues explore the possible role of CHD6 on the cellular response to oxidative damage. They observe that CHD6 levels are stabilized upon increasing oxidative environment and that CHD6 is actively recruited to sites of oxidative damage in a PAR-dependent manner. In fact this recruitment is, at least, partially dependent on the double chromodomain of CHD6 (interestingly CHD6 double chromodomain binds PAR) and independent of the

SSB repair scaffolding protein XRCC1. Although Δ CHD6 cells accumulate higher levels of DSBs in the presence of H₂O₂, when studying the repair efficiency of different DNA lesions no significant repair defect is detected. Curiously, CHD6 ablation promotes higher levels of ROS and oxidative damage-induced PAR. Authors find an explanation with the correlation between CHD6 absence and the downregulation of cellular antioxidant regulators. On the other hand, CHD6 regulates globally chromatin compaction resulting in ATM hyperactivation in the presence of low number of DSBs and increased G2/M checkpoint arrest in the presence of oxidative damage. Overall, CHD6 ablation confers sensitivity to oxidative damage. This is a very wide characterisation of the role of CHD6 on the cellular response to oxidative DNA damage. There are many interesting observations and most of conclusions are properly supported by experimental data. However, some key questions remain unanswered and the final model seems too speculative. **We also thank this reviewer for their support and interest in our work!**

Major comments

1. It has been shown by many labs that PAR-dependent recruitment of SSB repair factors (including XRCC1) occurs within few seconds. That is much quicker than the recruitment of CHD6 shown in Figure 2 and is in agreement with an XRCC1-independent recruitment of CHD6. However, since parylation occurs so quick, and since CHD6 binds PAR, why is CHD6 not recruited as quick as other PAR-binding factors? And why does it remain for so long? This result would support an indirect role of PAR in CDH6 recruitment or even a retention rather a recruitment one. Authors should clarify or discuss further this fact. FRAP experiments might help.

Our new analysis of the seven new truncation mutants of CHD6 has addressed many of the concerns raised here and clarified this issue (see Fig 1I and Fig 3). Close examination of CHD6 dynamics reveals there is an early and rapid (PAR-dependent) recruitment similar to XRCC1 and PARP1. This then progresses into a slower accumulation phase that lasts up to 8 min, and is distinct to XRCC1 and PARP1 which start to disperse in a steady fashion within only a few minutes of being recruited. By contrast, CHD6 is more or less stably retained up to about 16 min, then undergoes a steady dispersal. As explained above, we have mapped the PAR-dependent recruitment region to a small N-terminal region that is highly conserved with a very strong PAR-binding consensus motif contained within. We now suggest that (based on the data with our truncation mutants) that this region is responsible for the fast and early recruitment requiring PAR, similar to XRCC1 and PARP1. The double chromodomain and, to a lesser extent, the central core regions are then required for the slower recruitment and retention phases of CHD6 relocalization to DNA damage sites, which likely involves direct nucleosome binding and/or interactions with histone tails or DNA.

2. Although laser tracking looks convincing all experiments in Figure 2 should be quantified. A representative image of a single cell is not informative enough when comparing knockdowns (Figure 2D). In the case of Δ Chromo recruitment (Figure 2F), Control plot seems to be the same one than in Figure 2B. If that's the case authors should clarify if both experiments were performed at the same time. Since no biological replicates are shown and laser tracks usually exhibits broad experimental

variation controls should have been performed in parallel. If those tracks were not done at the same time, the experiment should be repeated and a proper control included.

All laser data is now quantified (this is now in Figure 3, with full statistical analysis shown within the Table embedded in Figure S1). With regards to our controls, indeed the reviewer is correct that the same plot was shown for all the original figures. The way we do this is that every time any mutant is being tested, we *always* first check a control (wildtype or untreated) cell that is subjected to microirradiation to confirm that on that specific day/experiment the system is working and the “expected” wildtype kinetics are observed. Thus new plots for WT/control cell kinetics are collected across the many months (indeed years now) that all our laser data has been obtained. As microirradiation experiments often require many runs to assemble to ~30 x >20 min live cell time courses needed for each mutant or condition, this can span a significant period of time. We have thus aggregated all of our wildtype / control laser into a single data set applied to every single experiment in Figure 3. This accounts for the varied time between experiments, and reflects the internal controls performed each time. We added a section to the methods to indicate this to the reader.

3. Due to the high ratio SSB:DSB induced by ROS is quite surprising that authors do not include SSB measurement but only DSBs markers such as 53BP1 and γ H2AX foci. DSB observed in laser tracks are probably the consequence of high power used and therefore coincident SSBs. In addition, Figure 1H suggests DSB-independent recruitment of CHD6. A rational quantification of SSB formation and repair in the absence of CHD6 should be included in this work. Alkaline comet assays are likely the best technical approach for that purpose. Could differences in SSBs explain the earlier activation of ATM and G2/M checkpoint rather the chromatin compaction status?

Great point! We now have added alkaline comet assay data (Figure 5B-D) to monitor SSB induction and resolution. As expected, we observe a subtle increase in SSB induction in the CHD6 deleted mutant. There was no difference in SSB repair, however, when examining the fraction of damage remaining (Fig 5D). This fits with all out previous observations. Given the subtle differences in SSB induction, similar to the DSB differences, we still do not think this can entirely explain the early ATM activation phenotype. However, one possible alternative explanation (to the chromatin hypothesis) is a direct activation of ATM via oxidative stress – indeed, there is evidence to support this from the Paull group. We have added this to our discussion.

4. Since FM-HCR is a plasmid-based DNA repair assay, it does not seem to be the best method for measuring chromatin remodellers dependent-repair. Authors could have used CHD1-4 knockdowns as positive controls. Is true that complementary data for studying DSB repair is shown but there is not supportive data in the case of MMR, NER and BER. Again, SSB induction and repair should be checked.

We have added in new controls using siRNA to deplete CHD2 and Ku80 – see Figure 5G, lower panels. We observe the anticipated increase in HR and reduction in NHEJ thought to be associated with CHD2 loss based on previous work from some of our team (HvA lab). Hence, this methods is at least somewhat sensitive to CHD enzyme status. As suggested, the SSB data is now included with the

comet data, backing up these observations. Further to this point, we strongly suspect that our FM-HCR plasmids are 'chromatinized' in vivo once transfected into cells, and the Nagel Lab (part of our team) are working towards a robust characterization of this (beyond the scope of this current study).

5. To confirm the role of CHD6 in the increase of oxidative stress through HMOX1 and TXNRD1 downregulation authors should actually show that CHD6 is relocalised to these genes upon oxidative stress. They should also try to complement DCF assay (Figure 3E) with CHD6 catalytic mutant and overexpressing NRF2.

Please see our comments to reviewer 1 with regards to CHIP (major point #6). With regards to the latter suggestion, we have been able to complement the DCF assay with wildtype CHD6 and present this data in figure 4E. This required us to re-clone CHD6 into an mPLUM tagged construct, as all GFP constructs interfered with the green channel required to monitor the DCF in the FACS machine. This cloning took some time, and we have not been able to obtain the catalytic dead mutants within the time frame of the revisions. However, given that the wildtype successfully complemented the H2O2 induced phenotype of CHD6-deleted cells in this assay, we hope the reviewer agrees that this is sufficient for now.

With regards to an NRF2 add-back, we instead were able to over-express HMOX1 (a target of NRF2 activity) and examine whether it could compensate for the absence of CHD6 using the PAR induction assay outlined in Figure 4 – this was in response to Reviewer #1's comments, but to a large extent it also addresses your request here. We hope this suffices, as (again) the available HMOX1 (and indeed other NRF2 substrate) constructs available to us were all GFP-tagged and thus not compatible with the DCF assay.

Minor comments:

1. In general, when performing statistic analysis non-significant differences should be indicated as well as significant ones are. Lack of these indications makes reader understand that some important pairs have not been tested. **We have added in "ns" to most relevant pairwise comparisons. In some cases, we have not done this simply for clarity and space reasons. We are happy to add in more if the reviewer finds any that they feel are critical are missing.**

2. Authors should quantify changes in the levels of other CHD proteins, especially in CHD2 blot, that seems overexposed, making not obvious that no significant changes occur. Figure 1B.
Done. This is now Fig 1B-C.

3. Authors should indicate either in the figure legend or in experimental procedures what is the control gene/RNA they use for relative quantification in all RT-PCR experiments.
We used GAPDH and also a control for TBP. We have added more detail about this to the methods.

4. Authors associate CHD6 levels with oxidative stress. Interestingly they show later that CHD6 is only recruited to ROS-induced DNA lesions. How are CHD6 levels regulated upon a non-oxidative source of DNA damage?

To address this we treated cells with etoposide, a topoisomerase poison that does not redox cycle. We observed no increase in CHD6 levels, although the induction of a DNA damage response was confirmed by immunoblotting for p53 levels and phosphorylation at serine 15. This data is in Fig 1E. Please note that we over-exposed this blot (compared to others in Figure 1) in order to see CHD6 at all in these cells.

5. Figure 2F. Imperfect overlap between CHD6 and 53BP1 seems anecdotal and not very convincing. **Agreed, and we removed this.**

6. Add stats to figure 3E. **Done.**

7. The statistic test used in Figure 2B should be indicated in figure legend, and be applied to Figure 2F. **A full table of statistical analysis of all laser data is now presented in Fig S11.**

8. Line 125. “via a potentially methylated PAR-binding partner” seems speculative. **Removed.**

9. Proliferation experiments are not very informative and could be moved to supplementary. In special Figure 6B, in which Control cell number saturates after 4 days. **We would prefer to keep this in the main text, as these experiments explain why our (10 day) clonogenic data with acute peroxide exposure lacks significant differences. We agree at day 4, both untreated cell lines reach saturation – however, what is important is what is occurring to the peroxide treated cells, which are both still within linear growth. We have altered the description of the results section to reflect this.** Authors should include statistic analysis (two-way ANOVA) on cell survival. **Done.**

10. Authors should discuss possible link between progressive ataxia in CHD6 Δ exon12 mice and SSB repair. Since there is an increased PAR induction in the absence of CHD6, do authors detect depleted NAD⁺ levels? This possibility is quite relevant and should be at least discussed. In fact, the second time authors cite reference 19 (line 52) that citation should be substituted by a reference to McKinnon or Caldecott labs where this SSB repair defects-dependent ataxias have been broadly documented. **Unfortunately, we have not been able to perform an NAD⁺ analysis in the timeframe of revisions, but have included a discussion of this in the manuscript as suggested. We have added the appropriate references as suggested also.**

11. Figure 1E. Endogenous CHD6. Square marking region selected for expanded image is moved. **Fixed.**

12. Sections in results are incorrectly numbered. **Our bad! Corrected.**

13. Line 234. is referred to Figure 5A bottom panel rather than 5B. **Fixed.**

Reviewers' comments:

Reviewer #1 (Remarks to the Author):

The authors have put considerable effort into responding to the comments of the reviewers and the paper is improved. However, some comments were not addressed experimentally (see below), partly for technical reasons. The key novelty remains the observation that CHD6 plays a role in the cells response to oxidative stress. The demonstration that HMOX1 rescues the increased PAR is an excellent experiment and suggests that CHD6 functions transcriptionally during oxidative stress. It would have strengthened this argument if HMOX1 could rescue cell survival following H₂O₂ (e.g. added to figure 7), ATM signaling and MNase sensitivity (likely not altered by HMOX1 re-expression) or other cell/molecular events altered in CHD6 deficient cells. An important experiment suggested by both reviewers was to ChIP for CHD6 at the promoters of potential target genes. While I can appreciate that this is technical challenging, it would have been useful to use e.g. RNA-seq (to get a broader view of transcriptional repression) or DNase I hypersensitivity (to check for compaction/transcriptional repression) at promoters of HMOX1 etc. Overall, the paper is still a little unclear on whether CHD6 exerts its effects through: (i) direct action at DSB; (ii) a general effect of loss of CHD6 leading to more "open" chromatin, which amplifies damage signaling; (iii) CHD6 is required for transcriptional activation of oxidative stress response proteins. Creating a CHD6 which lacks recruitment to damaged sites (e.g. mutations in the PAR domain) but which can still transcriptionally regulate HMOX1 etc would have helped with dissecting these functions. Overall, the paper presents an initial preliminary analysis of CHD6 functions during ox stress, but leaves many of the key mechanistic questions unanswered (define PAR domain; discriminate between repair and transcription functions; define impact of CHD6 on ox stress gene regulation).

(1) The model in figure 7 is overly complex and needs to be simplified and more adequately discussed in the discussion.

(2) p10, last paragraph. The statement "Relative to controls, the clonogenic potential of A549dCHD6 was reduced 10 days after acute H₂O₂ exposure, albeit not significantly" needs rewording. If the differences are not significant, then its not reduced.

(3) With regard to why CHD6 overexpression suppresses PAR – this has not really been addressed. It could be that overexpression of CHD6 increases chromatin compaction (check with MNase assay), which limits chromatin parylation. CHD6 stabilization during ox stress may then also lead to increased compaction, limiting DNA damage.

Reviewer #2 (Remarks to the Author):

Authors have done a real effort in answering most of my questions. New experiments and text corrections make manuscript look more solid and convincing. However, there are some key missing points that would require further modifications. At this point and considering the high interest of the observations shown and the enormous amount of work, indicated text corrections would be enough for this referee to support the acceptance of this manuscript in Nature Communications.

Related to previous review:

Major:

2. I could not find a section in methods describing how laser tracks are quantified.

3. Regarding differences in ATM activation, a small but significant difference in SSB formation exists (Figure 5B-C). This difference is probably enough to differential activation of ATM according to previous studies such as Dianov's lab PNAS. This should be included as a possibility in the

discussion as the disulfide cross-linkage within ATM dimers driven by elevated ROS levels has been. Authors should relax their conclusions in lanes 258-261 and 290-292.

4. Controls in FM-HCR don't look very convincing apart from the exacerbated HR in Ku80 knockdown. Is the NHEJ defect shown in siCHD2 statistically significant? SSB and DSB repair are backed up with repair kinetics but results about NER, MMR and abasic BER are not definitive.

5. Suggesting a role of CHD6 in transcriptional regulation of Nrf2-responding genes (Figure 7E.4) requires more evidence than shown in this manuscript. Suppression experiment with overexpressed HMOX1 is a very nice genetic evidence but still CHD6 recruitment upon H₂O₂ treatment would be required. ChIP experiments are essential for that purpose. The lack of a good antibody is a common issue when performing ChIP analysis and I acknowledge author's efforts with GFP-tag. However, GFP-trap is not the most appropriate strategy due to high background that overexpressed GFP usually show in ChIP experiments. Flag tag is way more useful for that aim. Anyway, if authors cannot provide this piece of evidence relaxation in their conclusions would be ideal. As an example, adding a question mark in their model (Figure 7E.4), using "suggest" rather than "confirm" in lane 211 and include in discussion section that further experiments would be required to confirm this role.

Minor:

3. I could not find a section in methods describing which is the control gene for RT-PCR.

I believe authors have done a big effort in fitting all panels into a quadrangular-profile figure, but a more-logically-organized distribution of sections is desirable.

Rebuttal Comments in **BLUE FOR REVIEWER #1** and in **ORANGE FOR REVIEWER #2**

Reviewer #1 (Remarks to the Author): The authors have put considerable effort into responding to the comments of the reviewers and the paper is improved. However, some comments were not addressed experimentally (see below), partly for technical reasons. The key novelty remains the observation that CHD6 plays a role in the cells response to oxidative stress. **We thank the reviewer for their continued support for our work, and recognition of effort in responding to concerns and comments.**

1. The demonstration that HMOX1 rescues the increased PAR is an excellent experiment and suggests that CHD6 functions transcriptionally during oxidative stress. It would have strengthened this argument if HMOX1 could rescue cell survival following H₂O₂ (e.g. added to figure 7), ATM signaling and MNase sensitivity (likely not altered by HMOX1 re-expression) or other cell/molecular events altered in CHD6 deficient cells. **We now provide new experimental evidence to address one of these suggestions. We have been able to successfully complement CHD6-deleted cells with HMOX1 and monitor cell growth/death over several days in the presence or absence of chronic oxidative stress. This data, now in Panel 7E, strengthens the argument (as suggested by the reviewer) that CHD6 functions transcriptionally during oxidative stress, as HMOX1 was able to rescue the majority of observable cell death/growth failure triggered by chronic hydrogen peroxide addition to media. Addressing the other (very good) suggestions represent much more substantial technical hurdles, and will take beyond a reasonable timeframe to address properly.**
2. An important experiment suggested by both reviewers was to ChIP for CHD6 at the promoters of potential target genes. While I can appreciate that this is technical challenging, it would have been useful to use e.g. RNA-seq (to get a broader view of transcriptional repression) or DNase I hypersensitivity (to check for compaction/transcriptional repression) at promoters of HMOX1 etc. **We have continued to pursue this, taking forward advice from both reviewers and abandoning GFP-tagged CHD6 expression constructs for ChIP in favour of FLAG-tagged constructs. We also reached out to the Walsh group, who have previously performed CHD6 ChIP using home-made polyclonal antibodies, for a small sample of this reagent. Unfortunately, we received no responses to our requests – we speculate that (as this was a polyclonal) it may be an exhausted resource but, as we received no replies, cannot say. Experiments with FLAG-CHD6 and ChIP in combination with cellular peroxide exposure remain extremely technically challenging, and progress is too slow for a reasonable timeframe for these revisions. While we have future plans for RNA-sequencing under several oxidative stress conditions with multiple new CHD6-deletion lines (in RPE1, 1BRhTERT ‘normal’ cells) currently being prepared, we feel that this is also beyond a reasonable expectation for the first description of CHD6 in the oxidative DNA damage response – particularly given the large datasets already shown and negligible space for further data in the manuscript – and hope you can agree.**
3. Overall, the paper is still a little unclear on whether CHD6 exerts its effects through: (i) direct action at DSB; (ii) a general effect of loss of CHD6 leading to more “open” chromatin, which amplifies damage signaling; (iii) CHD6 is required for transcriptional activation of oxidative stress response proteins. **We agree, and have expanded upon this in our discussion section and, indeed, have adapted the reviewers own very well-articulated comments into those points. The new experimental data with HMOX1 add-back strengthens the argument for the third possibility, and we have also highlighted this in our revised discussion.**

4. Creating a CHD6 which lacks recruitment to damaged sites (e.g. mutations in the PAR domain) but which can still transcriptionally regulate HMOX1 etc would have helped with dissecting these functions. **We are taking a residue-by-residue approach to mutagenesis within the short (~50aa) PAR-dependent binding region we have already identified in the existing data set for this study. This is also taking time and will be beyond the reasonable scope of this study, as it really requires a more rigorous structural understanding of the region (in complex with PAR) to do “properly”. This would be a paper in and of itself, as we hope you can appreciate.**
5. Overall, the paper presents an initial preliminary analysis of CHD6 functions during ox stress, but leaves many of the key mechanistic questions unanswered (define PAR domain; discriminate between repair and transcription functions; define impact of CHD6 on ox stress gene regulation). **Our new data adds to the notion that the main role for CHD6 in the oxidative DNA damage response (at least in terms of cell fate and DNA damage induction / PAR signalling) is through the regulation of antioxidant enzyme pathways. We make no assertion that CHD6 is involved in DNA repair directly, indeed we show convincing evidence of no role in HR, NHEJ and SSBR via multiple assays. This is a first description, and so we hope the reviewer will appreciate that there will be mechanistic questions raised by our work, while at the same time having identified (and in some cases consolidated to a significant extent) multiple roles for CHD6 in the DNA damage response.**
6. The model in figure 7 is overly complex and needs to be simplified and more adequately discussed in the discussion. **We agree, have split this up into multiple panels to streamline the layout, and have simplified the content. We have expanded the discussion to fit with the different possibilities, and have re-worded the figure legend to clearly describe our model.**
7. p10, last paragraph. The statement “Relative to controls, the clonogenic potential of A549dCHD6 was reduced 10 days after acute H₂O₂ exposure, albeit not significantly” needs rewording. If the differences are not significant, then its not reduced. **We agree and have re worded this to simply say “trended downwards but without statistical significance” and hope that is acceptable. For the narrative of the paper, it was useful to lead with a statement like this, as immediately we show that the lack of significance in the clonogenic is explained by the cell growth experiment, wherein significant cell death occurs much earlier than the relevant time point of a clonogenic.**
8. With regard to why CHD6 overexpression suppresses PAR – this has not really been addressed. It could be that overexpression of CHD6 increases chromatin compaction (check with MNase assay), which limits chromatin parylation. CHD6 stabilization during ox stress may then also lead to increased compaction, limiting DNA damage. **Thanks for this idea, and we have thought very carefully about it. Our new data reinforces the notion that this phenomena is most likely via CHD6-dependent signalling to antioxidant genes such as HMOX1, as this not only restores normal PAR induction, but mostly rescues the cell death phenotype of CHD6-deleted cells undergoing oxidative stress. Consequently, we favor a model where the suppression of PAR via CHD6 over-expression is mainly via the role of CHD6 in the antioxidant response, although agree we cannot say this with 100% certainty at this stage. With regards to the hypothesis raised by the reviewer, we note that PARP and PARylation have been documented to be important contributors to compacted (hetero)chromatin building by several groups (PMID 18676401, 20371698 and reviewed in 23731385) and/or PARP enzyme/activities are not excluded from such regions (PMID 23355608). From this**

literature, increased nucleosome compaction (at least as observed in interphase) is not a major block to PARylation of histones, indeed, it can be a consequence of it. Our supplementary data are also consistent with no role for CHD6 in the displacement (or return) of histone H1 via PARylation – indicating that the expression status of CHD6 does not influence PAR-dependent H1 dynamics (and any PARylated H1-related alterations to compaction / nucleosome accessibility) during a DNA damage response. We and others have also shown that oxidative DNA damage (such as IR) produces rapid and ATM-dependent chromatin *relaxation* observable via MNase assay and other methods (PMID 16862143, 18657500, 21642969, 25533843). Based on those data, and our data indicating the ATM-independence of CHD6 functions monitored so far (relocalization, etc.), it would be largely inconsistent to place CHD6 as a major chromatin compactor during a normal DNA damage response (as broad compaction has not been documented), or that such an activity explains any modulation of PAR signalling. Hence, we are reluctant to pursue that line of discussion and hope the reviewer agrees that this makes sense at this time.

Reviewer #2 (Remarks to the Author): Authors have done a real effort in answering most of my questions. New experiments and text corrections make manuscript look more solid and convincing. However, there are some key missing points that would require further modifications. At this point and considering the high interest of the observations shown and the enormous amount of work, indicated text corrections would be enough for this referee to support the acceptance of this manuscript in Nature Communications. **We thank this reviewer for their support and favorable assessment of our earlier revisions.**

1. I could not find a section in methods describing how laser tracks are quantified. **We have added a much more detailed methods section outlining how this was done.**
2. Regarding differences in ATM activation, a small but significant difference in SSB formation exists (Figure 5B-C). This difference is probably enough to differential activation of ATM according to previous studies such as Dianov's lab PNAS. This should be included as a possibility in the discussion as the disulfide cross-linkage within ATM dimers driven by elevated ROS levels has been. Authors should relax their conclusions in lanes 258-261 and 290-292. **Great suggestion and we have added a new discussion point to outline this possibility, together with a reference to that PNAS paper. We have also "relaxed" the wording of the stated sections.**
3. Controls in FM-HCR don't look very convincing apart from the exacerbated HR in Ku80 knockdown. Is the NHEJ defect shown in siCHD2 statistically significant? SSB and DSB repair are backed up with repair kinetics but results about NER, MMR and abasic BER are not definitive. **We have actually redone this FM-HCR experiment. The work in the revised manuscript used cells examined 72h post transfection with siRNA. In the revised work, we used cells monitored 96h post siRNA, and obtained much better knockdowns with results that were much more robust and similar between experimental repeats. In the new data, both CHD2 and Ku siRNA produced statistically significant reductions in NHEJ, fitting with the literature and serving as excellent positive controls for our work here. As the abasic BER assay for FM-HCR intrinsically involves key SSB steps, we argue that this endpoint is also supported (to a significant extent) by our alkaline comet assay results. We agree that the details of the different modalities of BER, as well as NER and MMR, will require future consolidation, and have added a line to indicate this to the results section describing this experiment.**

4. Suggesting a role of CHD6 in transcriptional regulation of Nrf2-responding genes (Figure 7E.4) requires more evidence than shown in this manuscript. Suppression experiment with overexpressed HMOX1 is a very nice genetic evidence but still CHD6 recruitment upon H₂O₂ treatment would be required. CHIP experiments are essential for that purpose. The lack of a good antibody is a common issue when performing CHIP analysis and I acknowledge author's efforts with GFP-tag. However, GFP-trap is not the most appropriate strategy due to high background that overexpressed GFP usually show in CHIP experiments. Flag tag is way more useful for that aim. Anyway, if authors cannot provide this piece of evidence relaxation in their conclusions would be ideal. As an example, adding a question mark in their model (Figure 7E.4), using "suggest" rather than "confirm" in lane 211 and include in discussion section that further experiments would be required to confirm this role. **Please see our response to Reviewer #1's second point. Also, as suggested here, we have altered our wording in the indicated lines (and other sections) to relax how we describe our conclusions. Question marks have also been included in the model to make clear what is still speculative.**
5. I could not find a section in methods describing which is the control gene for RT-PCR. **We used GAPDH, and had this in the original qPCR table, but admittedly didn't explicitly explain this. We have now added a much more detailed methods section outlining how this was done.**
6. I believe authors have done a big effort in fitting all panels into a quadrangular-profile figure, but a more-logically-organized distribution of sections is desirable. **We have re-arranged Figures 2, 5 and 7 where a logical left-right / top-down modality of sections was not originally adhered to 100%. All figures now (as best possible) have the A, B, C... etc. oriented either top-to-bottom then left-right OR left-right then top-bottom.**

REVIEWERS' COMMENTS:

Reviewer #1 (Remarks to the Author):

The authors have done a good job replying to the comments and the revised manuscript is greatly improved. The experiment showing rescue of CHD6 depletion by HMOX1 is an important addition to the experimental proof. While some areas remain to be clarified, including the relative contribution of CHD6 to transcription vs DNA repair, the paper is novel and reveals new functions for CHD6. Overall, it is now acceptable for publication.

Reviewer #2 (Remarks to the Author):

Authors have addressed most of my comments. I recommend this manuscript to be accepted and published in Nature Communications.